# A multi-modal transformer for predicting global minimum adsorption energy

Junwu Chen [1,2,6], Xu Huang [1,3,6], Cheng Hua [4], Yulian He [3,5] ✉ & Philippe Schwaller [1,2] ✉

The fast assessment of the global minimum adsorption energy (GMAE) between catalyst surfaces and adsorbates is crucial for large-scale catalyst screening. However, multiple adsorption sites and numerous possible adsorption configurations for each surface/adsorbate combination make it prohibitively expensive to calculate the GMAE through density functional theory (DFT). Thus, we designed a multi-modal transformer called AdsMT to rapidly predict the GMAE based on surface graphs and adsorbate feature vectors without site-binding information. The AdsMT model effectively captures the intricate relationships between adsorbates and surface atoms through the cross-attention mechanism, hence avoiding the enumeration of adsorption configurations. Three diverse benchmark datasets were introduced, providing a foundation for further research on the challenging GMAE prediction task. Our AdsMT framework demonstrates excellent performance by adopting the tailored graph encoder and transfer learning, achieving mean absolute errors of 0.09, 0.14, and 0.39 eV, respectively. Beyond GMAE prediction, AdsMT's cross-attention scores showcase the interpretable potential to identify the most energetically favorable adsorption sites. Additionally, uncertainty quantification was integrated into our models to enhance the trustworthiness of the predictions.

The adsorption energy of an adsorbate on the catalyst surface is crucial for determining the reactivity and selectivity of catalytic reactions. The highest catalytic activity of a material frequently resides at the optimal adsorption energy of the key reaction intermediates, according to the Sabatier principle[1–4]. Therefore, developing cheap and efficient adsorption energy evaluation methods is key for accelerating catalyst discovery. Currently, high-throughput screening of catalysts relies heavily on computationally expensive simulations like density functional theory (DFT)[5–8]. However, multiple adsorption sites and variable adsorbate geometries lead to numerous possible adsorption configurations and local minima on the potential

energy surface[9–11]. The local adsorption energy, which strongly depends on the initial structure of the simulation, might not well represent the catalytic activity. Several methods, including global optimization algorithms[12,13] and "brute-force" searches[14,15], have been employed to find the most stable adsorption structures and corresponding global minimum adsorption energies (GMAE). However, the high computational cost of such DFT-based methodologies inevitably imposes limitations on their large-scale implementation, given the immense catalyst design space.

Recent developments in machine learning (ML) algorithms hold great promises in approximating DFT-level accuracy with significantly

[1]Laboratory of Artificial Chemical Intelligence (LIAC), Institute of Chemical Sciences and Engineering, Ecole Polytechnique Fédérale de Lausanne (EPFL), Lausanne, Switzerland. [2]National Centre of Competence in Research (NCCR) Catalysis, Ecole Polytechnique Fédérale de Lausanne (EPFL), Lausanne, Switzerland. [3]Department of Chemistry and Chemical Engineering, Shanghai Jiao Tong University, Shanghai, China. [4]Antai College of Economics and Management, Shanghai Jiao Tong University, Shanghai, China. [5]University of Michigan-Shanghai Jiao Tong University Joint Institute (UM-SJTU JI), Shanghai, China. [6]These authors contributed equally: Junwu Chen, Xu Huang. ✉e-mail: yulian.he@sjtu.edu.cn; philippe.schwaller@epfl.ch

higher efficiency and lower computational costs[16–19]. Various ML models, such as random forests, multilayer perceptions, and graph neural networks (GNNs), have been explored to predict the adsorption energy of adsorbate-surface systems[11,20–26]. However, several drawbacks are present in most models, which (1) can only predict local minimum adsorption energies, (2) require specific binding information between the adsorbates and catalyst surfaces, and (3) exhibit poor generalizability limited to specific adsorbates. Recently, Ulissi et al. proposed the AdsorbML workflow[9], which combines heuristic search and ML potentials to accelerate the GMAE calculation. The ML potentials trained on the huge Open Catalyst 2020 (OC20) dataset achieve promising prediction accuracy and substantial speedups over DFT computations[9]. Moreover, Margraf et al.[10] proposed a global optimization protocol that employs on-the-fly ML potentials trained on iteratively DFT calculations to search the most stable adsorption structures. This method is versatile for various combinations of surfaces and adsorbates, significantly reducing DFT calculations as well as the reliance on prior expertise[10]. Despite notably mitigating computational expenses relative to DFT methods, these approaches still require the exploration of a large number of initial adsorption structures and iterative calculations to obtain the GMAE values.

Recently, multi-modal learning has become a research hotspot through the extraction and alignment of rich information from heterogeneous modalities for scientific research[27–31]. Among them, the multi-modal transformers demonstrate exceptional learning capability by associating different modalities with a cross-attention mechanism[30–34]. For instance, Kim et al.[30] created a multi-modal pre-training transformer that integrates atom-wise graphs and energy-grid embeddings to predict the properties of metal-organic frameworks (MOFs). Moreover, a prompt-guided multi-modal transformer proposed by Park et al.[31] demonstrated excellent performance in predicting the density of states (DOS) through modalities of graph embedding and energy-level embedding of crystals.

Herein, we propose a multi-modal transformer model, named AdsMT, which incorporates catalyst surface graphs and adsorbate feature vectors as heterogeneous input modalities to directly predict the GMAE of diverse adsorption systems without the acquisition of any site-binding information. The AdsMT is designed to capture the intricate relationships between adsorbates and the multiple adsorption sites on surfaces through the cross-attention mechanism, thereby avoiding the enumeration of adsorption configurations. As illustrated in Fig. 1a, three GMAE datasets comprising diverse catalyst surfaces and adsorbates were introduced for the challenging GMAE prediction task. Our AdsMT demonstrates excellent performance in predicting GMAE, with mean absolute errors (MAE) below 0.15 eV for two of the datasets. A transfer learning strategy was also proposed to further improve AdsMT's performance on small-sized datasets. Moreover, cross-attention weights are exploited to identify the most energetically favorable adsorption sites and demonstrate the interpretable potential of AdsMT. The calibrated uncertainty estimation is integrated into our AdsMT for reliable GMAE prediction. Overall, AdsMT exhibits strong learning ability, generalizability, and interpretable potential, making it a powerful tool for fast GMAE calculations and catalyst screening.

## Results
### AdsMT architecture
AdsMT is a multi-modal Transformer that takes periodic graph representations of catalyst surfaces and feature vectors of adsorbates as inputs to predict the GMAE of each surface/adsorbate combination without any site binding information. As depicted in Fig. 1b, the AdsMT architecture consists of three parts: a graph encoder $E_G$, a vector encoder $E_V$, and a cross-modal encoder $E_C$. In the graph encoder, the unit cell structure of each catalyst surface is modeled as a graph $\mathcal{G}$ with periodic invariance by self-connecting edges and radius-based edge construction (see Methods for details). The atom-wise embeddings of surfaces are output from the graph encoder and passed into the cross-

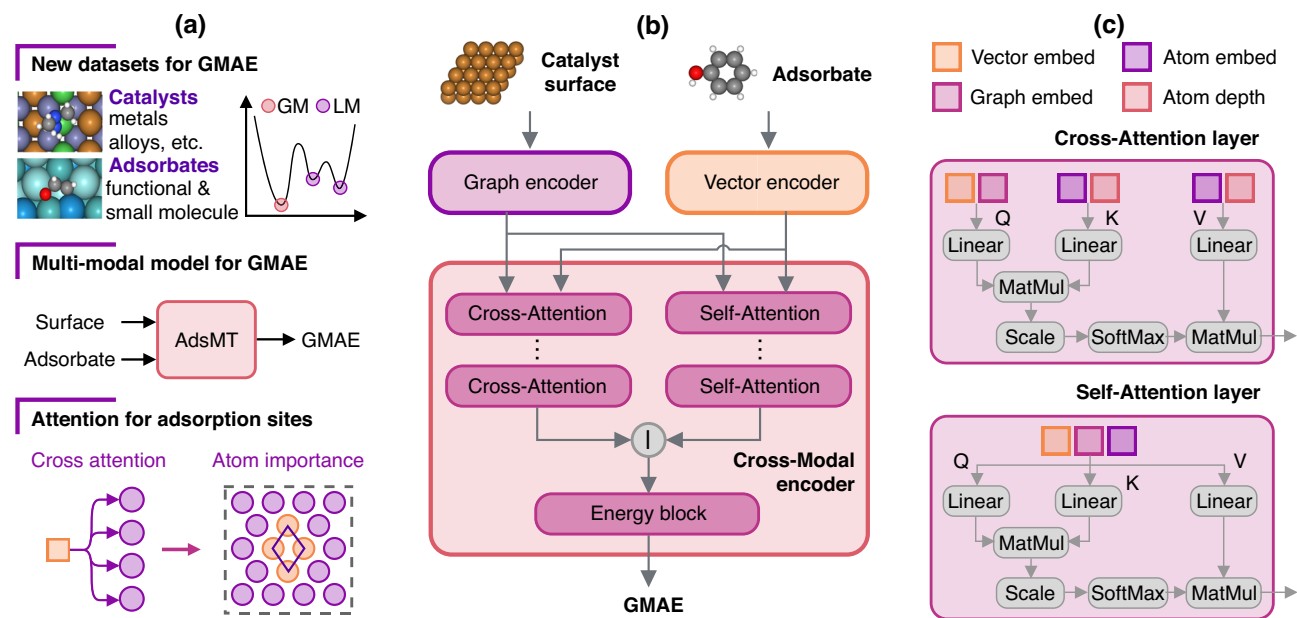

**Fig. 1 | Overall schematics and architecture of AdsMT. a** The schematic overview of this study. We present three datasets containing diverse combination of catalysts and adsorbates for predicting the global minimum adsorption energy (GMAE). The upper-right plot illustrates the difference between global minima (GM) and local minima (LM). AdsMT is a multi-modal model that processes separate surface and adsorbate inputs to predict GMAE. **b** The architecture of AdsMT. AdsMT consists of three blocks: a graph encoder for catalyst surface encoding, a vector encoder for adsorbate encoding, and a cross-modal encoder for GMAE prediction from embeddings of surfaces and adsorbates. **c** Illustration of cross-attention and self-attention layers in the cross-modal encoder. In the first cross-attention layer, the concatenated adsorbate vector embeddings and surface graph embeddings form the query matrix (Q), while the concatenated atomic embeddings and depth embeddings serve as the key (K) and value (V) matrices. Each atomic depth vector encodes the relative position of an atom within the surface (e.g., top-layer or bottom-layer). In the self-attention layer, the stacked atom embeddings, surface graph embeddings, and adsorbate vector embeddings are used as the input Q, K, and V.

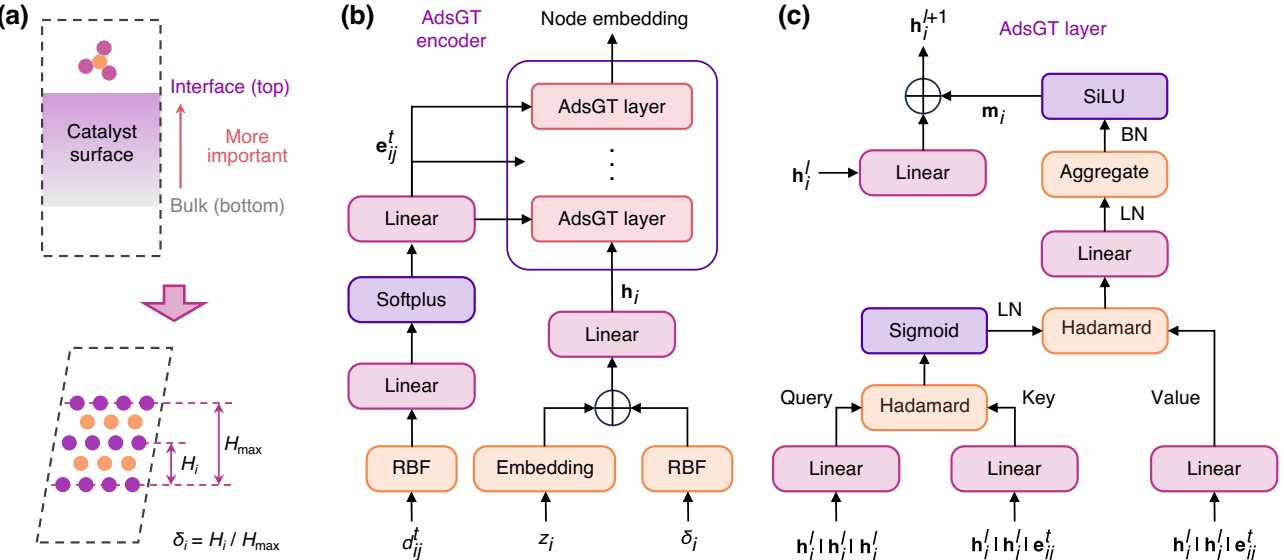

**Fig. 2 | Schematics and architecture of AdsGT encoder designed for surface graphs. a** The positional encoding in AdsGT encoder. AdsGT encoder uses positional encoding to represent the relative height of each atom on the surface, where importance increases from bottom to top. Each atom $i$ is assigned a positional feature $\delta_i = H_i/H_{max}$, where $H_i$ denotes the atom's height and $H_{max}$ is the maximum height in the surface. **b** The architecture of AdsGT encoder. AdsGT encoder includes radial basis function (RBF) expansions, embeddings, and graph attention layers. **c** Illustration of each AdsGT layer with an edge-wise attention mechanism delineated by three steps: calculation of edge-wise attention coefficients, edge-wise message calculation, and node update. $d_{ij}^t$ and $e_{ij}^t$ are the distance and embedding of $t$-th edge between atom $i$ and $j$. $z_i$ is the atomic number of atom $i$. $h_i^l$ is the atomic embedding of atom $i$ at $l$-th AdsGT layer. LN and BN represent layer normalization and batch normalization, respectively.

modal encoder. Any geometric graph neural network, such as SchNet[35] and GemNet[36], can serve as the graph encoder in the AdsMT framework. For the vector encoder, molecular descriptors are chosen to represent adsorbates (see Methods for details), while multilayer perceptron (MLP) is used to compute vector embeddings from adsorbate descriptors and passed to the cross-modal encoder.

The cross-modal encoder takes atomic embeddings of surfaces and vector embeddings of adsorbates as inputs to predict the GMAE. It comprises cross-attention layer(s), self-attention layer(s), and an energy block (Fig. 1b, c). The adsorption energy primarily arises from the interaction between the catalyst surface and the adsorbate, while the resulting surface atomic displacements also influence it[37–39]. Therefore, the cross-attention layer is assigned to capture the complex relationships between the adsorbate and all surface atoms, while the self-attention layer is expected to learn the interactions between atoms within the surface caused by adsorption (e.g., atomic displacements). In the first cross-attention layer (Fig. 1c), the concatenated matrix of adsorbate vector embeddings and surface graph embeddings is employed as the query matrix, while the concatenated matrix of atomic embeddings and depth embeddings serves as the key and value matrices. Each atomic depth vector describes the relative position (e.g., top-layer or bottom-layer) of an atom within the surface (Methods). In the self-attention layer, the stacked matrix of atom embeddings, surface graph embeddings, and adsorbate vector embeddings are set to the input query, key, and value. The aggregated output of the final self-attention layer is concatenated with the output of the last cross-attention layer, and passed into the energy block to predict the GMAE. The detailed algorithm of the cross-modal encoder is described in the Methods.

The graph encoder employed in the AdsMT model plays a pivotal role in capturing the structural and chemical features of catalyst surfaces. Unfortunately, existing GNNs fail to discriminate between top-layer and bottom-layer atoms when representing a surface as a graph. Practically, only the top-layer atoms of surfaces are capable of interacting with adsorbates, rendering them inherently more important than other atoms in terms of adsorption energy. Therefore, we

designed a graph transformer called AdsGT specifically for encoding surface graphs. As depicted in Fig. 2a, a positional encoding method is proposed to compute the positional feature $\delta_i$ for each atom based on fractional height relative to the underlying atomic plane. This approach augments the model's understanding of surface structures and differentiates between top and bottom layer atoms. Figure 2b shows the architecture of the AdsGT encoder, which consists of radial basis function (RBF) expansions, embeddings, and graph attention layers. Different from the conventional graph transformer like Graphormer[40], the AdsGT layer (Fig. 2c) adopts an edge-wise attention mechanism, delineated by three sequential steps: edge-wise attention coefficients calculation, edge-wise message calculation, and node update. More details about AdsGT architecture and its positional encoding are described in the Methods.

## GMAE benchmark datasets

We introduced three GMAE benchmark datasets named OCD-GMAE, Alloy-GMAE and FG-GMAE from OC20-Dense[9], Catalysis Hub[7], and 'functional groups' (FG)-dataset[25] datasets through strict data cleaning (see Methods for details), and each data point represents a unique combination of catalyst surface and adsorbate. As shown in Fig. 3a and Supplementary Tables 1–4, three GMAE datasets possess different sizes and span diverse ranges of chemical space. Alloy-GMAE comprises 11260 combinations (largest), covering 1916 bimetallic alloy surfaces and 12 small adsorbates of less than 5 atoms (*O, *NH, *CH$_2$, etc.). FG-GMAE exhibits a medium scale with 3308 combinations, featuring 202 adsorbates with diverse functional groups (e.g., alcohols, amidines, aromatics) alongside only 14 pure metal surfaces. OCD-GMAE consists of 973 combinations spanning 967 inorganic surfaces (intermetallics, ionic compounds, etc.), coupled with 74 adsorbates (O/H, C$_1$, C$_2$, N-based). As illustrated in Fig. 3c and Supplementary Figs. 1–3, the catalyst surfaces within OCD-GMAE showcase the most diverse elemental composition (54 elements), including alkali/alkaline earth metals, transition/post-transition metals, metalloids, and reactive nonmetals. Comparatively, the surfaces of Alloy-GMAE involve 37 species of transition/post-transition metals, while FG-GMAE only

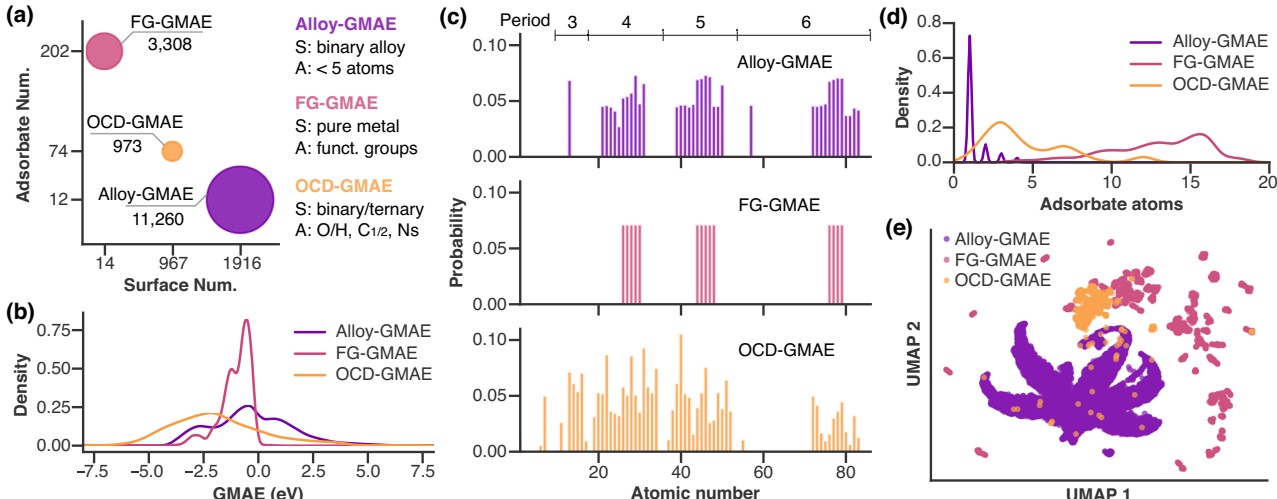

**Fig. 3 | Statistical analysis and comparison of three datasets on global minimum adsorption energy (GMAE). a** The overview and comparison of three GMAE datasets, with details on dataset size, unique types of surfaces and adsorbates, and brief descriptions of surfaces (S) and adsorbates (A). **b** Distribution of the GMAE target in three datasets. **c** The occurrence probability of elements for surface compositions in GMAE datasets. **d** Distribution of number of atoms of adsorbates in GMAE datasets. **e** The uniform manifold approximation and projection (UMAP) plot of surface/adsorbate combinations in GMAE datasets, where each combination is depicted by the smooth overlap of atomic positions (SOAP) descriptors of surfaces and RDKit descriptors of adsorbates. This panel illustrates that three GMAE datasets represent distinct chemical spaces, with some areas of overlap.

has 14 transition metals. Regarding GMAE values (Fig. 3b), FG-GMAE ranges from −4.0 to 0.8 eV, primarily concentrated around −0.9 eV. In comparison, Alloy-GMAE and OCD-GMAE present a broader spectrum of GMAE values (−4.3 - 9.1 and −8.0 - 6.4 eV), and encompass more surface/adsorbate combinations with positive GMAEs. Moreover, the Uniform Manifold Approximation and Projection (UMAP) algorithm was utilized to visualize the chemical space of three GMAE datasets on a two-dimensional plane (Supplementary Note 2), where the distances between each surface/adsorbate combination are correlated with the differences in feature space[41,42]. Each surface/adsorbate combination is depicted by the smooth overlap of atomic positions (SOAP) descriptors[43,44] of surfaces and RDKit descriptors[45] of adsorbates. Figure 3e demonstrates that three GMAE datasets delineate separate chemical spaces, albeit with certain overlapping regions.

## AdsMT performance and transfer learning

The prediction performance of the AdsMT framework with different graph encoders (AdsGT, CGCNN[46], SchNet[35], DimeNet++[47], GemNet-OC[48], ET[49] and eSCN[50]) were evaluated on the three GMAE datasets (Fig. 4a and Supplementary Table 6–7). All outcomes stem from ten repeated experiments with different random seeds following an 8:1:1 training/validation/testing set ratio (Methods). In addition to MAE, we adopt a new evaluation metric termed success rate (SR), representing the percentage of predicted energies within 0.1 eV of the DFT-computed GMAEs[9]. On the Alloy-GMAE dataset, the MAE values of AdsMT models range from 0.14 to 0.17 eV, with lower MAE tending towards higher SR. The proposed AdsGT surpasses other graph encoders and achieves the best MAE (0.143 eV) and SR (66.3%) in GMAE prediction. While on the smaller-sized FG-GMAE dataset, AdsMT models yield even lower MAE ( ~0.1 eV) and higher SR ( ~69%), with the best MAE of 0.095 eV and an SR of 71.9% through employing the AdsGT encoder. The fewer element types of surfaces and narrower GMAE distribution of the FG-GMAE dataset could be beneficial to the GMAE prediction task. Notably, our AdsMT outperforms the GAME-Net model (MAE = 0.18 eV) that was specifically designed for FG-Dataset and required site binding information[25]. More challenging dataset splits based on surface or adsorbate type were tested on the Alloy-GMAE and FG-GMAE datasets to explore the generalization performance of AdsMT to unseen surfaces or adsorbates (Supplementary Tables 15–18). For surface- or adsorbate-

based data partitioning, a set of unique types was obtained, of which 80% types were randomly sampled for training, and each 10% types were used for validation and testing, respectively. Therefore, the types of surfaces or adsorbates present in the test set are not included in the training and validation sets. As shown in Supplementary Tables 15 and 16, using surface-based data split leads to an increase of approximately 0.02 eV in the MAE with a corresponding decrease of around 6% in the success rate compared to random split. Although the prediction accuracy slightly decreases under surface-based data partitioning, the best MAE and success rate of AdsMT model on Alloy-GMAE were 0.158 eV and 60.1%, respectively. Similarly, as presented in Supplementary Tables 17 and 18, adsorbate-based data partitioning results in an increase of approximately 0.04 eV in MAE and a reduction of about 8% in success rate compared to random split, and AdsMT achieves the best MAE of 0.123 eV and the best success rate of 65.3% on the FG-GMAE. The slight accuracy drops demonstrate the robust generalization capability of AdsMT to unseen surfaces or adsorbates.

In contrast, all AdsMT models exhibit unsatisfactory performance on the OCD-GMAE dataset, with MAE exceeding 0.5 eV and SR below 15%. The underperformance could be attributed to the limited dataset size (<1000) and the intricate composition of catalyst surfaces involving 54 elements. Nevertheless, the AdsMT model with AdsGT encoder still achieves the best MAE of 0.571 eV and an SR of 13.5% on the OCD-GMAE dataset, outperforming other graph encoders. In addition, the Uni-Mol+ model (49M)[51] pre-trained on the huge OC20 dataset[52] was explored for GMAE prediction by initial structure sampling, which only achieved a success rate of about 32% and highlighted the difficulty of the OCD-GMAE dataset (Supplementary Table 19).

To enhance the AdsMT performance under data scarcity, we implemented a transfer learning strategy that entails pre-training on data with local minimum adsorption energy (LMAE). To this end, we established OC20-LMAE, a dataset comprising 363,937 surface/adsorbate combinations alongside their LMAEs, derived through data cleaning of the OC20 dataset[52] (Methods). It should be noted that both OCD-GMAE and OC20-LMAE datasets originate from the Open Catalyst Project[52,53] with analogous surface and adsorbate types, which will be advantageous for transfer learning. As illustrated in Fig. 4b, each AdsMT model undergoes initial pre-training on the OC20-LMAE, followed by fine-tuning on the GMAE datasets while selectively freezing

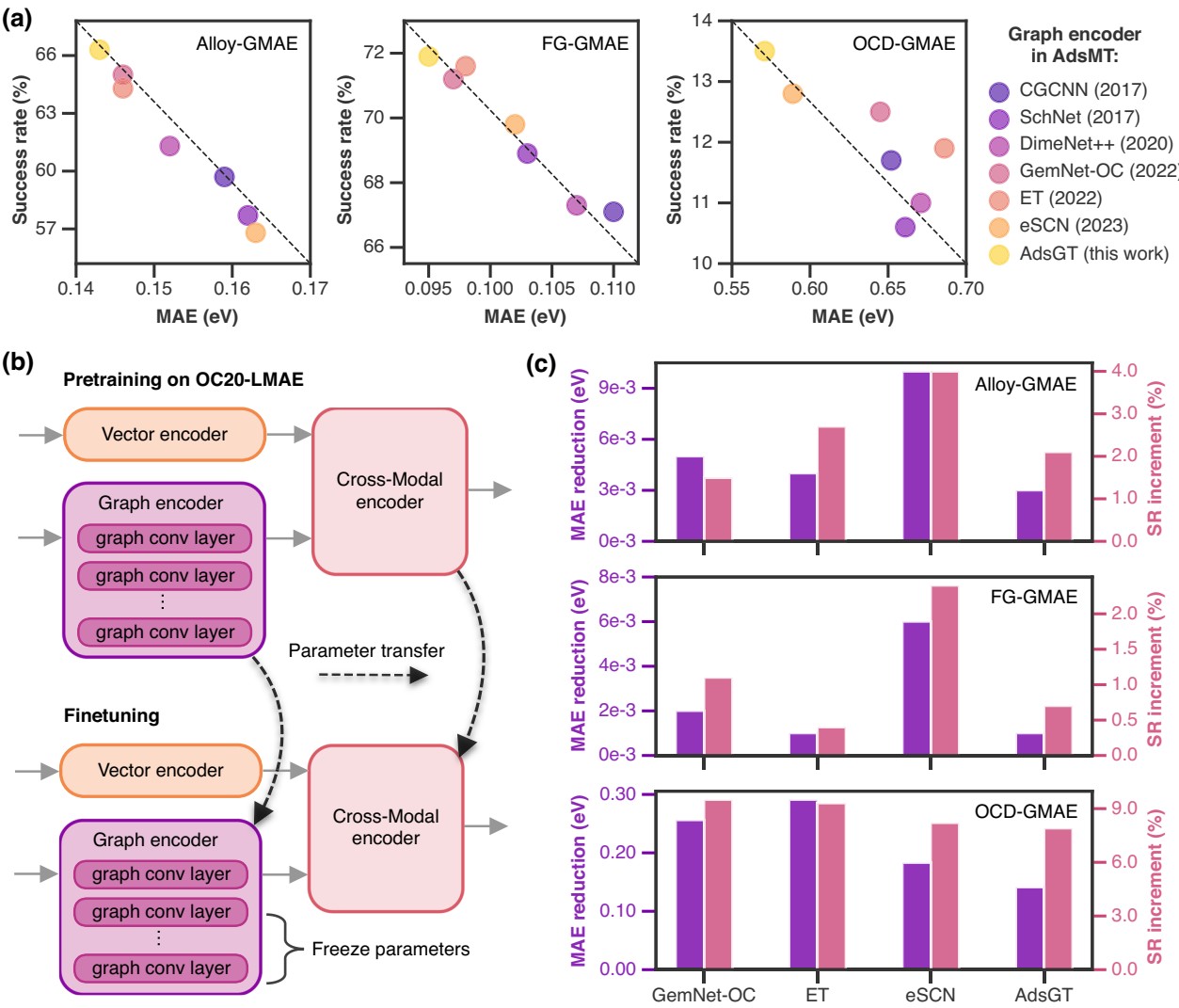

**Fig. 4 | AdsMT performance on global minimum adsorption energy (GMAE) prediction and transfer learning. a** The prediction success rate (SR) and mean absolute error (MAE) of AdsMT models with different graph encoders[35,46–50] on the three GMAE datasets. **b** Schematic illustration of our transfer learning strategy.

Each AdsMT model is pre-trained on the OC20-local minimum adsorption energy (LMAE) dataset, and then fine-tuned on a GMAE dataset with the graph encoder parameters selectively frozen. **c** AdsMT's performance gains after transfer learning using different graph encoders[48–50].

graph encoder parameters. The efficacy of our transfer learning strategy is elucidated in Fig. 4c and Supplementary Tables 8–10, where AdsGT and GNNs reported in the past two years (refs.[48–50]) are chosen as the graph encoders for AdsMT. On the OCD-GMAE dataset, AdsMT models achieve obvious performance gains after transfer learning, resulting in all MAE reductions surpassing 0.14 eV and SR increments exceeding 7%. Particularly, the ET encoder enables AdsMT to achieve an MAE reduction of 0.291 eV and a 9.3% increase in SR, and the GemNet-OC encoder facilitates AdsMT to attain an MAE reduction of 0.256 eV and a 9.5% increase in SR. The best performance of AdsMT on the OCD-GMAE was obtained after transfer learning, yielding an MAE of 0.389 eV and a SR of 22.0%. On the contrary, transfer learning only provides slight improvements for AdsMT models on the Alloy-GMAE and FG-GMAE, likely attributable to substantial dissimilarities in catalyst surface types between these datasets and OC20-LMAE (Supplementary Note 9)[54,55]. The effectiveness of transfer learning mainly depends on the quality of pre-training data and the similarity between the source and target domains. It is important to note the inherent difference between LMAE and GMAE. We randomly sampled 100 surface-adsorbate combinations from OC20-LMAE and calculated their GMAE using the DFT method and AdsorbML pipeline[9]. The mean

absolute difference between GMAE and LMAE is about 0.27 eV. The MAEs of the AdsMT models without transfer learning on the OCD-GMAE dataset are about 0.6 eV. Therefore, it is reasonable that pre-training on LMAE data can improve the AdsMT performance on OCD-GMAE, despite the inherent difference between LMAE and GMAE. However, AdsMT already exhibits commendable predictive performance on the FG-GMAE and Alloy-GMAE datasets, with MAE values below or proximal to 0.1 eV. The inherent errors (>0.2 eV) in LMAE data could hinder improving the AdsMT performance on both datasets through transfer learning.

Overall, these results demonstrate the remarkable ability of the AdsMT framework to rapidly predict GMAE of diverse surface/adsorbate combinations without any binding information. Utilizing separate information of catalysts and adsorbates as input, the AdsMT could generalize to predictions for unseen surfaces or adsorbates, making it suitable for efficient virtual screening of catalysts where adsorption structures are rarely available.

**Adsorption site identification from cross-attention**
Beyond predicting adsorption energies, identifying adsorption sites holds particular importance in catalyst design and reaction mechanism

studies[24,56,57]. In this context, we explored the application of attention scores from cross-attention layers to estimate the most energetically favorable adsorption sites on catalyst surfaces[58]. As illustrated in Fig. 5a, the average cross-attention score of each surface atom with

respect to the adsorbate is computed from all attention heads of the last cross-attention layer, which implies the relative importance of each surface atom in adsorbate binding[58]. The surface atom(s) with the highest average cross-attention score is hypothesized as the most

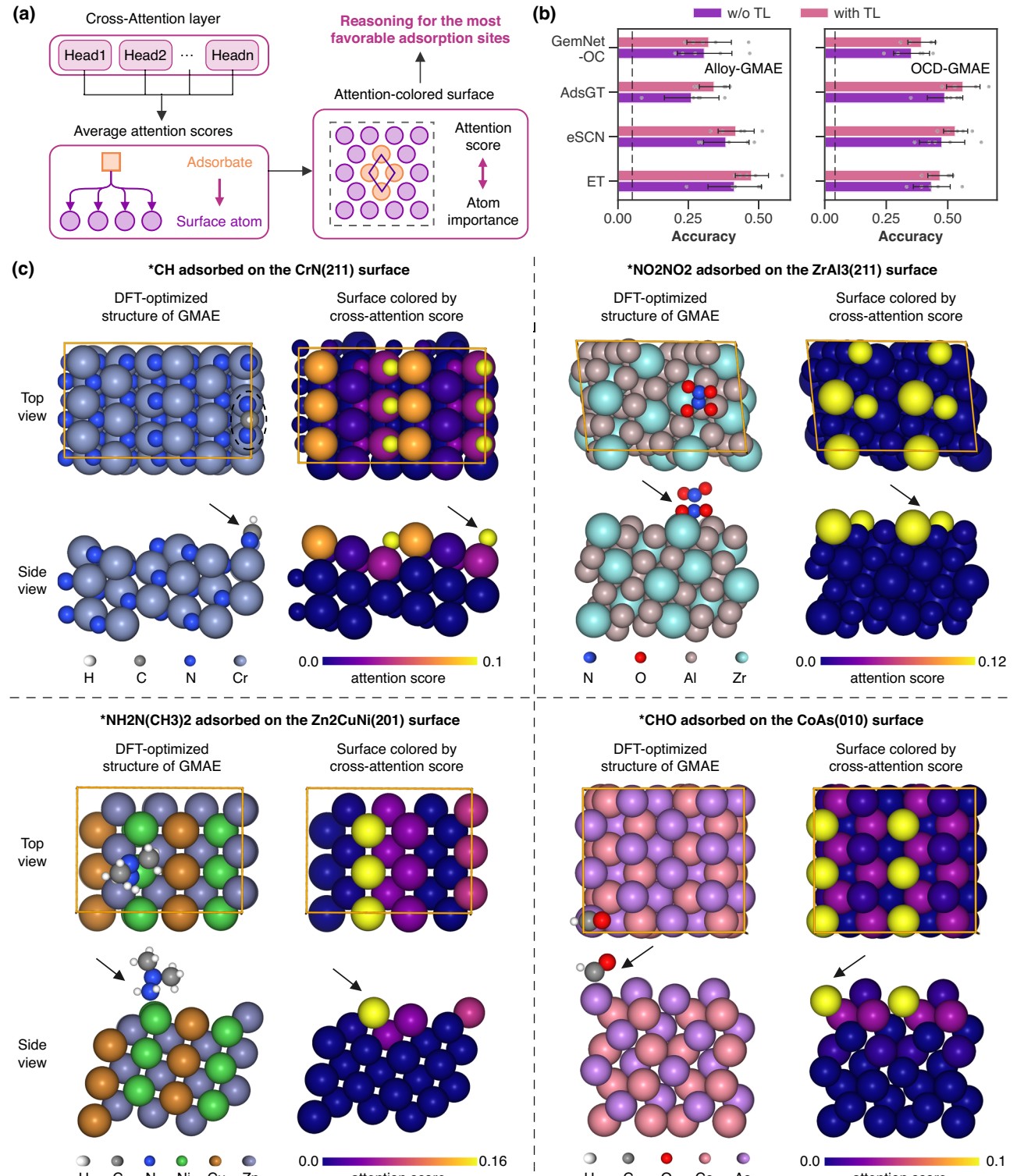

**Fig. 5 | Adsorption site identification by cross-attention scores. a** Schematic of identifying the most energetically favorable adsorption sites from the average cross-attention score of each surface atom relative to the adsorbate, which is calculated over all attention heads in the last cross-attention layer. **b** The accuracy ($n = 5$) of AdsMT models adopting different graph encoders[48–50] in identifying optimal adsorption sites with or without transfer learning (TL). The black dotted lines represent the accuracy of random atom selection. The error bars represent standard deviations from five experiments. **c** Four examples of the comparison between (left) global minimum adsorption structures optimized by density functional theory (DFT) and (right) attention score-colored surfaces computed by the trained AdsMT model with AdsGT encoder. The black arrows point to the most stable adsorption sites.

favorable adsorption site. To assess the reliability of adsorption site identification from cross-attention scores, no information regarding adsorption structures or sites was provided to the AdsMT model during training. The trained AdsMT model is employed to suggest the optimal adsorption site for each surface/adsorbate combination and compared with the ground truth from DFT calculations.

Figure 5c presents four examples contrasting cross-attention score-colored surfaces (right) with DFT-optimized adsorption configurations under GMAE (left). For the combination of CrN(211) surface and *CH adsorbate, six equivalent N atoms of the surface possess much higher cross-attention scores compared to other atoms, and the adsorbate *CH is bonded with two of these N atoms in the DFT-optimized GMAE structure. For $*NH_2N(CH_3)_2$ on the $Zn_2CuNi(201)$, three equivalent Ni atoms in the top layer of the surface show the highest cross-attention scores, while the adsorbate binds to one of them in the GMAE structure. In addition, our model effectively distinguishes between top-layer and sub-layer atoms, benefiting from incorporating atomic depth embeddings in the cross-attention layers. These tendencies are consistent with other random examples in the Alloy-GMAE and OCD-GMAE datasets (Supplementary Fig. 8–11), indicating that the most favorable adsorption sites strongly relate to the atoms with high cross-attention scores. However, the AdsMT model is unsuitable for reasoning about adsorption sites on simple monometallic surfaces (e.g., FG-GMAE dataset), where the top-layer atoms are completely equivalent and have identical attention scores. Furthermore, we computed the accuracy of AdsMT models in identifying optimal adsorption sites on the Alloy-GMAE and OCD-GMAE (Supplementary Note 3). As illustrated in Fig. 5b, the AdsMT models demonstrate commendable identification capabilities for optimal adsorption sites across both datasets, substantially surpassing the accuracy obtained through random atom selection (black dotted line). The AdsMT model adopting the ET encoder achieves the highest accuracy of 0.48 on the Alloy-GMAE dataset, while the AdsMT model with the AdsGT encoder exhibits the highest accuracy of 0.56 on the OCD-GMAE. The implementation of transfer learning was also found to improve the AdsMT's accuracy for adsorption site identification. In addition, the cross-attention scores also have the potential to identify the different types of adsorption sites (e.g., top, bridge, hollow) through the improved method (Supplementary Note 10).

These results confirm that our AdsMT architecture can indeed learn the complex association between adsorbates and surface atoms through the cross-attention mechanism, underscoring its interpretable potential. The trained AdsMT model can be a valuable tool to rapidly identify energetically favorable adsorption sites of a specific adsorbate on the surface.

### Calibrated uncertainty estimation

From the practical perspective of virtual catalyst screening, it is desirable that the models can provide uncertainty estimation for their predictions, enabling researchers to evaluate the reliability of predictions and assign experimental effort more efficiently. To this end, an ensemble of independent AdsMT replicates is trained to estimate the uncertainty from the variance of individual models' predictions, which is a widely recognized method for effective uncertainty quantification (Methods)[24,59,60]. The AdsMT ensemble's predictions were ranked based on their uncertainty estimations (Supplementary Note 4), and the correlation between uncertainty and prediction MAE was investigated. As depicted in Supplementary Fig. 12a, AdsMT's predictions with lower uncertainty tend to have lower MAEs across the three GMAE datasets. Moreover, the Spearman correlation coefficients between the estimated uncertainty and prediction MAEs for the AdsMT models with different graph encoders consistently exceed 0.98 on the three GMAE datasets (Supplementary Fig. 12b). The results show that the AdsMT's estimated uncertainty is significantly correlated with the

predicted MAE, and its predictions are highly accurate at low uncertainty levels[24,60].

Furthermore, we investigated whether the AdsMT's uncertainty estimation is well-calibrated and statistically significant, thereby avoiding overconfidence or underconfidence[60-62]. Supplementary Fig. 13 presents the calibration curves and corresponding miscalibration areas of AdsMT models with different graph encoders on the three GMAE datasets (Supplementary Notes 5, 6), which is an effective approach to evaluating the calibration of uncertainty estimates[24,60,61]. It is notable that the calibration curves of AdsMT models closely approximate the ideal diagonal line and exhibit small miscalibration areas less than 0.1. The results prove that AdsMT's uncertainty estimations are well-calibrated and scaled with errors[60-62].

Overall, all results indicate that AdsMT's uncertainty estimation is reliable and well-calibrated. The precise uncertainty quantification is crucial for active learning and experimental validation, which can drive data expansion and candidate prioritization during the catalyst discovery.

## Discussion

We have presented AdsMT, a general multi-modal transformer framework for directly predicting GMAE of chemically diverse surface-adsorbate systems without relying on any binding information. The AdsMT integrates heterogeneous input modalities of surface graphs and adsorbate feature vectors, demonstrating excellent predictive performance on two GMAE benchmark datasets. Utilizing separate input information of catalysts and adsorbates, the AdsMT could generalize predictions for unseen surface/adsorbate combinations, making it suitable for efficient virtual screening of catalysts where adsorption structures are rarely available. Furthermore, the AdsMT is insensitive to surface geometric fluctuations without changes in atomic connectivity, which is advantageous for virtual screening across different materials/catalyst databases (Supplementary Note 11). Moreover, AdsMT achieves a speedup of approximately eight orders of magnitude compared to DFT calculations, and four orders of magnitude faster than machine learning interatomic potentials (MLIP) combined with heuristic search[9] (Supplementary Note 7). Such a high efficiency and low computational cost endow AdsMT with great promises for fast GMAE prediction and large-scale screening of catalysts.

In terms of data scarcity, AdsMT remains poised for enhancement, as indicated by its unsatisfactory performance on the OCD-GMAE dataset. It was shown that transfer learning is effective in addressing this challenge. In future work, MLIP can be employed to acquire coarse GMAE data for model pretraining, which is much cheaper than LMAE data from DFT calculations. Moreover, it can be particularly interesting to integrate AdsMT with active learning, as it enables the iterative expansion of the training datasets towards underexplored regions of catalyst space and improves the model's reliability.

The application of identifying the most advantageous adsorption sites from AdsMT's cross-attention scores is promising, despite the current accuracy is not high enough. An intriguing avenue for future research lies in incorporating domain knowledge such as adsorbate geometric information into model training, potentially enhancing the model's capability for GMAE prediction and adsorption site identification. Moreover, considering the surfacial atom importance for adsorption sites as a prediction target and fusing it into the loss function can be beneficial for the model to learn the complex relationship between catalyst surfaces and adsorbates.

Another natural extension to this work involves combining our AdsMT with MLIP and DFT calculations for catalyst screening in specific reactions (Supplementary Note 7). Each catalyst crystal can generate a large number of surface structures due to varying Miller indices and absolute positions of surface planes. Combined with uncertainty estimation, AdsMT can be used for rapid preliminary screening in huge catalyst libraries and pinpoint a small range of candidate catalyst

surfaces with desired GMAE and low uncertainty. Afterwards, more precise methods such as DFT can be used to further validate the top candidate catalysts. This strategy holds promise for significantly reducing computational costs while achieving reliable virtual catalyst screening.

## Methods

### GMAE benchmark datasets

Three GMAE datasets, named Alloy-GMAE, FG-GMAE and OCD-GMAE, were filtered from Catalysis Hub[7], 'functional groups' (FG)-dataset[25], and OC20-Dense[9] datasets, respectively. Each of the source datasets enumerated all adsorption sites on surfaces and performed DFT calculations on various possible adsorption configurations. The data cleaning was conducted to sort the local adsorption energies and take the lowest adsorption energy of all conformations as the GMAE target for each surface/adsorbate combination. Each data point in the datasets represents a unique combination of catalyst surface and adsorbate. Random splitting is adopted on three datasets during the model evaluation.

In addition, a similar data cleaning procedure was employed on the OC20 dataset[52] to create a new dataset named OC20-LMAE, which comprises surface/adsorbate pairings along with their local minimum adsorption energies (LMAE). The data points with anomalies (adsorbate desorption/dissociation, surface mismatch) are removed. The OC20-LMAE dataset contains 363,937 data points and serves as an effective resource for model pretraining. Specifically, its training set consists of 345,254 data points, while the validation set comprises 18,683 data points. Further detailed descriptions of the datasets are provided in the Supplementary Note 1.

### Surface graph

Each input catalyst surface is modeled as a graph $\mathcal{G}$ consisting of $n$ nodes (atoms) $\mathcal{V} = \{v_1, \ldots, v_n\}$ and $m$ edges (interactions) $\mathcal{E} = \{\epsilon_1, \ldots, \epsilon_m\} \subseteq \mathcal{V}^2$. $\mathbf{H} = [\mathbf{h}_1, \mathbf{h}_2, \cdots, \mathbf{h}_n]^T \in \mathbb{R}^{n \times k}$ is the node feature matrix, where $\mathbf{h}_i \in \mathbb{R}^k$ is the $k$-dimensional feature vector of atom $i$. $\mathbf{E} \in \mathbb{R}^{m \times k'}$ is the edge feature matrix, where $\mathbf{e}_{ij}^t \in \mathbb{R}^{k'}$ is the $k'$-dimensional feature vector of $t$-th edge between node $i$ and $j$. $\mathbf{X} = [\mathbf{x}_1, \mathbf{x}_2, \cdots, \mathbf{x}_n]^T \in \mathbb{R}^{n \times 3}$ is the position matrix, where $\mathbf{x}_i \in \mathbb{R}^3$ is the 3D Cartesian coordinate of atom $i$. For periodic boundary conditions (PBC), let the matrix $\mathbf{C} = [\mathbf{a}, \mathbf{b}, \mathbf{c}]^T \in \mathbb{R}^{3 \times 3}$ depicts how the unit cell is replicated in three directions $\mathbf{a}$, $\mathbf{b}$ and $\mathbf{c}$.

Ignoring periodic invariance will lead to different graph representations and energy predictions for the same surface[63]. Different from crystals, the presence of the vacuum layer breaks the periodicity along the direction perpendicular to the surface. This means that the catalyst surfaces exhibit periodicity only in the $\mathbf{a}$ and $\mathbf{b}$ directions. Thus, the infinite surface structure can be represented as

$$
\begin{aligned}
\hat{\mathbf{H}} &= \left\{ \hat{\mathbf{h}}_i | \hat{\mathbf{h}}_i = \mathbf{h}_i, i \in \mathbb{Z}, 1 \le i \le n \right\}, \\
\hat{\mathbf{X}} &= \left\{ \hat{\mathbf{x}}_i | \hat{\mathbf{x}}_i = \mathbf{x}_i + k_1 \mathbf{a} + k_2 \mathbf{b}, i, k_1, k_2 \in \mathbb{Z}, 1 \le i \le n \right\}.
\end{aligned}
\tag{1}
$$

To encode such periodic patterns, the infinite representation of the surface is used for graph construction, and all nodes and their repeated duplicates are considered to build edges. Given a cutoff radius $r_c \in \mathbb{R}$, if there is any integer pair $(k_1', k_2')$, such that the Euclidean distance $d_{ji} = \| \mathbf{x}_j + k_1' \mathbf{a} + k_2' \mathbf{b} - \mathbf{x}_i \|_2 \le r_c$, then an edge is constructed from $j$ to $i$ with the initial edge feature $d_{ji}$. It should be pointed out that self-loop edges ($i = j$) are also considered if there exists any integer pair $(k_1', k_2')$ other than $(0, 0)$ such that $d = \| k_1' \mathbf{a} + k_2' \mathbf{b} \|_2 \le r_c$.

### Adsorbate feature

The representation of adsorbate is crucial for models to predict the lowest adsorption energy for a given combination of surface and adsorbate. Some important adsorbates (e.g., *H, *O, *N) have only one atom, and message passing in GNNs cannot work for one node without

an edge. Many adsorbate species (e.g., $*CO_2$, *CO, *OH, $*NH_2$) consist of fewer than four atoms or two bonds, which makes capturing important chemical information difficult through atomic representation and graph learning. On the other hand, molecular descriptors based on expert knowledge can quickly and accurately capture the chemical information of adsorbates, especially for small adsorbates or new adsorbates without structural information. Therefore, molecular descriptors are used to represent adsorbates rather than the widely used molecular graphs. $\mathbf{P} = [\mathbf{p}_1, \mathbf{p}_2, \cdots, \mathbf{p}_s]^T \in \mathbb{R}^{s \times k''}$ is the adsorbate feature matrix, where $\mathbf{p}_c \in \mathbb{R}^{k''}$ is the $k''$-dimensional feature vector of the adsorbate for the surface/adsorbate combination $c$ ($1 \le c \le s$). In this study, the molecular descriptors of adsorbates were calculated by RDKit package[45], where $k'' = 208$ for adsorbate feature vectors.

### AdsGT graph encoder

**Positional feature.** Unlike molecular graphs, the importance of each atom in the catalyst surface differs for adsorption energy prediction (Fig. 2a). For example, atoms at the top layers are more important, while atoms at the bottom are less important. Moreover, GNNs are unable to determine the relative heights of atoms on a surface based on a surface graph, making it impossible to distinguish between top-layer and bottom-layer atoms. To help models understand the varying importance of atoms at different relative heights (Fig. 2a), each atom $i$ of a surface graph will get a positional feature $\delta_i$ computed by

$$
\delta_i = \frac{h - h_{min}}{h_{max} - h_{min}},
\tag{2}
$$

where $h$ is the height of the atom $i$ and calculated by the projection length of the atom coordinate $\mathbf{x}_i$ on the $\mathbf{c}$ vector. $h_{max}$ and $h_{min}$ represent the maximum and minimum heights of surface atoms, respectively. Specifically, $\delta_i = 1$ indicates that the atom $i$ is located at the topmost layer, while $\delta_i = 0$ means that the atom $i$ is located at the bottommost layer. Then, $\delta_i$ is expanded via a set of exponential normal radial basis functions $e^{RBF}$ to compute the positional embedding $\zeta_i$ of surface atom $i$:

$$
\zeta_i = e_k^{RBF}(\delta_i) = \exp\left(-\beta_k (\delta_i - \mu_k)^2\right),
\tag{3}
$$

where $\beta_k$ and $\mu_k$ are fixed parameters specifying the center and width of the radial basis function $k$, respectively.

In the initialization, atomic number $z_i$ is passed to the embedding layer and summed with the positional embedding $\zeta_i$ to compute the initial node embedding $\mathbf{h}_i^0$. The distance $d_{ij}^t$ of $t$-th edge between node $i$ and $j$ is expanded via a set of radial basis functions (RBF) and transformed by linear layers and Softplus activation function to obtain the edge embedding $\mathbf{e}_{ij}^t$. The message passing phase follows an edge-wise attention mechanism[63]. In the $l$-th ($0 \le l \le L$) attention layer, edge-wise attention weights $\boldsymbol{\alpha}_{ij}^t$ and message $\mathbf{m}_{ij}^t$ of $t$-th edge between node $i$ and $j$ are calculated based on $\mathbf{h}_i^l$, $\mathbf{h}_j^l$ and $\mathbf{e}_{ij}^t$ according to

$$
\mathbf{q}_{ij} = LN_Q^l\left(\mathbf{h}_i^l | \mathbf{h}_i^l | \mathbf{h}_i^l\right), \quad \mathbf{k}_{ij}^t = LN_K^l\left(\mathbf{h}_i^l | \mathbf{h}_j^l | \mathbf{e}_{ij}^t\right), \quad \mathbf{v}_{ij}^t = LN_V^l\left(\mathbf{h}_i^l | \mathbf{h}_j^l | \mathbf{e}_{ij}^t\right), \tag{4}
$$

$$
\boldsymbol{\alpha}_{ij}^t = \frac{\mathbf{q}_{ij} \circ \mathbf{k}_{ij}^t}{\sqrt{d_{\mathbf{k}_{ij}^t}}}, \quad \mathbf{m}_{ij}^t = \text{sigmoid}\left(\text{LNorm}\left(\boldsymbol{\alpha}_{ij}^t\right)\right) \circ \mathbf{v}_{ij}^t,
\tag{5}
$$

where $LN_Q^l$, $LN_K^l$ and $LN_V^l$ are three linear transformations, $\circ$ represent the Hadamard product, $|$ denotes concatenation, and LNorm is the layer normalization operation. Then, the message $\mathbf{m}_i$ of node $i$ from all neighbors $\mathcal{N}_i$ is computed by

$$
\mathbf{m}_i = \sum_{j \in \mathcal{N}_i} \sum_h \text{LNorm}\left(W_m^l \mathbf{m}_{ij}^t + b_m^l\right),
\tag{6}
$$

and the embedding of node $i$ is updated based on the message $\mathbf{m}_i$ according to

$$\mathbf{h}_i^{l+1} = W_u^l \mathbf{h}_i^l + b_u^l + \sigma\left(\mathrm{BNorm}(\mathbf{m}_i)\right), \tag{7}$$

where $W_m^l$ and $W_u^l$ are two learnable weight matrices, while $b_m^l$ and $b_u^l$ are two learnable bias vectors. $\sigma$ denotes the activation function, and BNorm represents batch normalization.

## AdsMT architecture

The proposed AdsMT model consists of three parts: a graph encoder $E_G$, a vector encoder $E_V$, and a cross-modal encoder $E_C$. Each surface/adsorbate combination $c$, consisting of a surface graph $\mathcal{G}_c$ and an adsorbate feature vector $p_c$, is defined as the model input, and the GMAE of the combination is set as the prediction target. Surface graphs and adsorbate feature vectors are passed to the graph encoder $E_G$ and the vector encoder $E_V$ for embedding learning, respectively. Then, both embeddings are passed to the cross-modal encoder $E_C$ for the cross-modal learning and GMAE prediction. The details of these parts are as follows.

**Graph encoder.** Prior to capturing the complex interaction between the surface graphs and the adsorbate features, geometric GNNs are used to encode the surface graphs into atom-wise embedding, which contains chemical and structural information. Formally, given a surface graph $\mathcal{G}_c = (\mathbf{H}, \mathbf{E})$ for the combination $c$, the atom embedding matrix $\mathbf{H}'$ is computed according to:

$$\mathbf{H}' = E_G(\mathbf{H}, \mathbf{E}) \in \mathbb{R}^{n \times k}, \tag{8}$$

whose $i$-th row indicates the representation of atom $i$. It is noteworthy that any geometric GNN, such as SchNet[35] and GemNet[36], can serve as the graph encoder in the AdsMT framework.

**Vector encoder.** A simple multilayer perceptron (MLP) is used to encode the feature vectors of adsorbates, and the adsorbate embedding of the combination $c$ is calculated based on

$$\mathbf{p}_c' = \mathrm{MLP}(\mathbf{p}_c), \tag{9}$$

**Cross-modal encoder.** The cross-modal encoder comprises a cross-attention module, a self-attention module, and an energy block. The cross-attention module is assigned to model the inter-modality and capture the complex relationships between the adsorbate and all surface atoms. Initially, the additional inputs of the cross-attention module are computed based on:

$$\mathbf{g}_c = \frac{1}{n}\sum_{i=1}^{n} \mathbf{h}_i', \quad \mathbf{H}' = \left[\mathbf{h}_1', \mathbf{h}_2', \cdots, \mathbf{h}_n'\right]^T \in \mathbb{R}^{n \times k}, \tag{10}$$

$$\mathbf{s}_i = W^S e^{\mathrm{RBF}}(\delta_i), \quad \mathbf{S} = \left[\mathbf{s}_1, \mathbf{s}_2, \cdots, \mathbf{s}_n\right]^T \in \mathbb{R}^{n \times k}, \tag{11}$$

where $\mathbf{g}_c$ is the surface graph embedding, $W^S$ is a learnable weight matrix, $\mathbf{s}_i$ is the depth embedding of surface atom $i$, and $\mathbf{S}$ is the surface atom depth embedding matrix similar to the position encoding of AdsGT encoder. The depth embedding $\mathbf{s}_i$ describes the relative position of atom $i$ in the surface (e.g., top layer, bottom layer). It could facilitate cross-attention layers to understand the surface structures and the importance of different atoms for adsorption. Then, each cross-attention layer is carried out as defined in the following equations:

$$\mathbf{a}_0 = (\mathbf{p}_c' \mid \mathbf{g}_c), \quad \mathbf{q}_l = \mathbf{a}_{l-1} W_l^Q, \tag{12}$$

$$\mathbf{K}_l = (\mathbf{H}' \mid \mathbf{S}) W_l^K, \quad \mathbf{V}_l = (\mathbf{H}' \mid \mathbf{S}) W_l^V, \tag{13}$$

$$\mathbf{a}_l = \mathrm{Cross\text{-}Attention}\left(\mathbf{q}_l, \mathbf{K}_l, \mathbf{V}_l\right) = \mathrm{softmax}\left(\frac{\mathbf{q}_l \mathbf{K}_l^T}{\sqrt{2k}}\right)\mathbf{V}_l, \tag{14}$$

where $l = 1, ..., L_1$ indicates the index of the cross-attention layers, and $W_l^Q$, $W_l^K$, $W_l^V$ are three learnable weight matrices. The final output $\mathbf{a}_{L_1}$ of the cross-attention module reflects the complex interaction between the surface atoms and the adsorbate.

Moreover, the self-attention module is designed to learn the interactions between atoms within the surface of the adsorbate caused by adsorption (e.g., atomic displacements). Initially, the stacked features $\mathbf{R}_0$ is computed by:

$$\mathbf{R}_0 = \left(\mathbf{H}', \mathbf{g}_c, \mathbf{p}_c'\right)^T. \tag{15}$$

Then, each self-attention layer is denoted as:

$$\mathbf{Q}_l' = \mathbf{R}_{l-1} W_l^{Q'}, \quad \mathbf{K}_l' = \mathbf{R}_{l-1} W_l^{K'}, \quad \mathbf{V}_l' = \mathbf{R}_{l-1} W_l^{V'}, \tag{16}$$

$$\mathbf{R}_l = \mathrm{Self\text{-}Attention}\left(\mathbf{Q}_l', \mathbf{K}_l', \mathbf{V}_l'\right) = \mathrm{softmax}\left(\frac{\mathbf{Q}_l'(\mathbf{K}_l')^T}{\sqrt{k}}\right)\mathbf{V}_l', \tag{17}$$

$$\mathbf{R}_l = \left(\mathbf{H}_l', \mathbf{g}_{c,l}, \mathbf{p}_{c,l}'\right)^T, \tag{18}$$

where $l = 1, ..., L_2$ indicates the index of the self-attention layers, and $W_l^{Q'}$, $W_l^{K'}$, $W_l^{V'}$ are three learnable weight matrices. The final output $\mathbf{z}$ of the self-attention module is calculated based on $\mathbf{R}_{L_2}$:

$$\mathbf{z} = \left(\mathbf{g}_{c,L_2} \mid \mathbf{p}_{c,L_2}'\right). \tag{19}$$

In the energy block, the multilayer perceptron (MLP) is used to compute the GMAE of the surface/adsorbate combination $c$ based on the final output of the cross-attention module $\mathbf{a}_{L_1}$ and the self-attention module $\mathbf{z}$:

$$y = \mathrm{MLP}\left(\mathbf{a}_{L_1} \mid \mathbf{z}\right). \tag{20}$$

## Model training

The training procedures have been executed by minimizing the MAE as the loss function using the AdamW optimizer[64]. The learning rate value is adjusted by the reduce-on-plateau scheduler. Each GMAE dataset underwent a random split into training, validation, and test sets with a ratio of 8:1:1. To scale the GMAE target, a standardization method was conducted using the mean and standard deviation of the GMAE values from the training set. The experiment results are derived from ten separate runs with different random seeds. Each model was trained on a single NVIDIA Tesla A100 GPU at float32 precision. To explore the benefits of transfer learning, models with and without transfer learning have the same architecture and hyperparameters. More training details including model hyperparameters are depicted in Supplementary Note 8.

## Implementation details

AdsMT models were built with PyTorch Geometric 2.2.0[65] running over PyTorch 1.13.1[66], with surface structures processing by Atomic Simulation Environment 3.22.1 package[67]. RDKit 2022.9.5 package[45] was used to generate the molecular descriptors for adsorbates. Matplotlib 3.7.2[68] and NGLview 3.0.8[69] were used to draw the plots presented in this work.

## Uncertainty quantification

The model ensemble method is used for uncertainty quantification, a technique widely acknowledged for its efficacy in uncertainty

estimation. Specifically, each of the ten AdsMT replicas shared identical architectures and hyperparameters, yet their learnable parameters were initialized with distinct random seeds. Denoting $\hat{y}_k(x_i)$ as the prediction from the $k$-th individual model for a given input surface/adsorbate combination $c_i$, AdsMT's final GMAE prediction $\mu(c_i)$ and its estimated uncertainty $\sigma(x_i)$, are derived from the mean and standard deviation of the individual model's predictions based on:

$$\mu(x_i) = \frac{1}{M}\sum_{k=1}^{M}\hat{y}_k(x_i), \quad \sigma(x_i)^2 = \frac{1}{M}\sum_{k=1}^{M}\left(\hat{y}_k(x_i) - \mu(x_i)\right)^2 \quad (21)$$

### Reporting summary

Further information on research design is available in the Nature Portfolio Reporting Summary linked to this article.

### Data availability

The datasets used in this study are available via Zenodo (https://doi.org/10.5281/zenodo.12104162)[70] and Figshare (10.6084/m9.figshare.25966573)[71]. This provides three GMAE benchmark datasets (Alloy-GMAE, FG-GMAE, OCD-GMAE) and OC20-LMAE dataset for model pretraining. Source data are provided with this paper.

### Code availability

The source code of the AdsMT framework is available on GitHub at https://github.com/schwallergroup/AdsMT (ref. 72).

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

## Acknowledgements

J.C. and P.S. acknowledge support from NCCR Catalysis (grant number 180544/225147), a National Centre of Competence in Research funded by the Swiss National Science Foundation. Y.H. would like to thank the National Natural Science Foundation of China (22202131) and the Shanghai Science and Technology Development Funds of the "Rising Star" Sailing Program (22YF1419400) for the financial support. C.H. would like to thank the National Natural Science Foundation of China (72301172 and 72394370:72394375) and the Shanghai Education Commission Chenguang Program (22CGA12) for the financial support.

## Author contributions

J.C. and X.H. contributed equally to this work. J.C. and X.H. contributed to methodology, model creation, writing, visualization and assessment. C.H. contributed to methodology, model design and writing. Y.H. and P.S. contributed to conceptualization, methodology, model creation, writing, assessment, funding and project supervision.

## Competing interests

The authors declare no competing interests.
