## [Transparent Peer Review file · Nature Communications]

A multi-modal transformer for predicting global minimum adsorption energy

Corresponding Author: Professor Philippe Schwaller

Version 0:

Reviewer comments:

Reviewer #1

(Remarks to the Author)

The paper presents a machine learning approach, AdsMT, for predicting the global minimum adsorption energy (GMAE) of catalyst surfaces. The proposed method uses a multi-modal transformer architecture that integrates surface graphs and adsorbate feature vectors through a cross-attention mechanism. This is a significant contribution. AdsMT predicts unseen surface-adsorbate combinations without binding site information. The method can also provide candidate adsorption sites based on the analysis of the attention weights. The integration of uncertainty quantification is also nice to experimentalists, as it provides a means of assessing the reliability of the model's predictions.

The flexibility of AdsMT is also noticeable. The three-part structure—graph encoder, vector encoder, and cross-modal encoder allows the choice of employing different graph encoders like SchNet, DimeNet++, and GemNet.

While the AdsMT framework's performance on datasets like FG-GMAE and Alloy-GMAE is impressive, the results for the OCD-GMAE dataset are less promising. The authors mention that transfer learning improves performance on smaller datasets, but this aspect could be explored more deeply. It would be beneficial to have a more detailed analysis of why the transfer learning strategy leads to only moderate improvements in some cases and how the model could be further refined to address these challenges.

The authors could also explore incorporating other forms of domain knowledge, such as geometric constraints or known adsorption site preferences, to improve the model's interpretability and accuracy.

Despite the need of further performance enhancement for being useful for experimentalists, the paper is a significant contribution and would be of interest to readers of Nat Commun

(Remarks on code availability)

Reviewer #2

(Remarks to the Author)

Chen et al. developed a transformer model to compute the global minimum adsorption energy of small adsorbates on metal (and metal alloy) surfaces. The model uses surface information (in the form of graphs), adsorbate information (using molecular descriptors), and multi-modal learning to compute the global minimum energy value for an adsorbate-surface pair. The resulting model offers some predictive capabilities in and of itself but with transfer learning from local minimum energy structures is able to have prediction errors between 0.1 – 0.4 eV. While these errors are still on the higher side, this model is quite useful from the standpoint of quick screening knowing only the surface information and basic adsorbate descriptors and not needing detailed adsorption structure knowledge. Indeed, the authors show that even using machine-learned potentials, computing global minimum adsorption structures can be time-consuming compared to using their model. For this reason, this work is interesting and worthy of publication in this journal, however, after consideration of the following questions and clarifications.

1. The authors choose multi-modal transformer learning and a specific type of self- and cross-attention model structure. It is

unclear why this technique is even necessary considering the limited size of the data set. Are there comparisons of model performance against some benchmarks (similar techniques that only predict global minimum energy structures? Or comparison with ML potentials, although it's not a perfect one-to-one comparison.

2. The authors correctly do not use structural information of adsorbates while building the adsorbate encoder but refer to RDKit-derived features. They should explain further what these features are and whether they need 3D structure information (and if yes, how do they get this). In a real-world application, the user of such a model may only hypothesize the species connectivity (e.g. they may be able to say the relevant intermediates are NH₂ or CH₂) but not be able to provide specific structures.

3. The surface graph seems to be generated from a periodic structure available from the OC database. Therefore, how sensitive are the predictions to the atom distances (or broadly speaking lattice constants)? A user may want to rely only on connectivity information (and approximate lattice constant information from a database such as ICSD, Materials project, AFLOWlib, etc.) for a new structure (not in the OC database). Ideally, therefore, the model should be insensitive (or only mildly sensitive) to atom-atom distances on the surface.

4. Using the attention scores to suggest which surface atoms are likely to participate in binding is interesting but given that the model never takes any information about adsorbate binding as input, this "interpretation" is probably just some kind of correlation that the model learns (e.g. whenever there are surface Cr atoms or N atoms, the binding energy may be high). In any case, this model is likely not able to say if the preferred adsorption mode is monodentate or bidentate, or if the site is a bridge, three-fold, or a top site, etc. In that sense, are the authors overinterpreting the capabilities of the model?

5. It was interesting to see that the model was able to learn from the local minimum structures. The naïve thinking would have been that such transfer learning would actually be detrimental because the LMAE dataset would have a lot of a one-to-many mapping (one adsorbate-surface combination having multiple energy values). Any thoughts on why this transfer learning works? Or does it work only because the encoding part of the model is learned using this pre-training step? Can the authors also clarify if there are adsorbate-surface combinations in the LMAE dataset that are not in the GMAE dataset?

(Remarks on code availability)

Reviewer #3

(Remarks to the Author)

The paper addresses an important problem in catalysis of finding the global minimum adsorption energy for an adsorbate on a catalyst's surface. They propose a technique that first computes a graph embedding for the catalyst and a vector embedding for the adsorbate. These are then combined with attention models to predict the energy.

Results are demonstrated on 3 different datasets with and without transfer learning. The results on dataset with lower diversity are reasonable with a success rate of 66-72%. However success on the OCD dataset which has a much larger variety of catalysts was only 14-17%.

The paper has two main issues that limit its potential to be published:

1) Since the success rates are low and the relaxed structures are not provided by the approach, how would the approach be used in practice? If relaxed structures are provided, DFT can be performed to double check their correctness. Without them, a full DFT relaxation would need to be performed to check the final results (assuming the adsorbate site is correctly found, otherwise it's even worse). Practically speaking if a DFT relaxation take 200 steps, an approach that produces relaxed structures gets 200 more "tries" than an approach that doesn't.

2) Since this paper emphasizes performance gains, other baselines should be considered. For instance, IS2RE models that directly predict relaxed energies from initial structures (no relaxation is performed) could be used along with sampling 2-10 random adsorbate positions. This would be marginally slower than the authors approach and may be more accurate. The reason for this is there exists a lot more data for training an IS2RE model. Training a model using the OC20 Dense dataset is very challenging due to its small size, which was originally intended as a testing dataset, not a training dataset.

Minor comments:

1) Are catalyst surface and adsorbate features really multi modal? In ML when I hear multi modal I usually think of images vs. text or similar differences. They are both atom representations, and it is unclear how much advantage is gained from treating them separately. For instance the adsorbate could be encoded using the mean atom embedding?

2) What are the eSCN results after transfer learning? From the bar graphs you can't determine whether it outperforms AdsGT.

3) How was "The data cleaning was conducted to remove abnormal adsorption structures" performed for the datasets? How many datapoints remained? This makes it hard to compare against previous papers.

4) The authors state "To the best of our knowledge, this is the first work that directly predicts the GMAE of diverse adsorption systems without the acquisition of any site-binding information." Please see AdsorbDiff: Adsorbate Placement via Conditional Denoising Diffusion, Kolluru and Kitchin, ICML 2024. They show a method that directly predicts the relaxed structure without having to perform relaxations.

5) Reword "We built three GMAE benchmark datasets named OCD-GMAE, Alloy-GMAE and FG GMAE". It'd be more

accurate to say you "filtered" three dataset.

(Remarks on code availability)

Version 1:

Reviewer comments:

Reviewer #1

(Remarks to the Author)

I think the authors addressed most of the concerns from the reviewers, and the paper is ready for publication from my point of view. However, I am not an expert on this energy calculation. As an experimentalist, my judgement is based on the method's usefulness for analyzing adsorption energy in a practical system. The opinion of the other two reviewers may be more important.

(Remarks on code availability)

I have not reviewed the code but checked its availability. Reviewing of this code is beyond my expertise.

Reviewer #2

(Remarks to the Author)

The authors have addressed the questions raised by this reviewer and is publishable in its modified form.

(Remarks on code availability)

Reviewer #3

(Remarks to the Author)

Thank you to the authors for providing a thorough response to the reviewer's comments.

After reading the responses, my main question remains whether the results on FG and Alloy make up for the results on OC20. The results on FG and Alloy demonstrate that with enough training data of similar surfaces and adsorbates, the AdsMT approach can be effective. The 8:1:1 split makes it very likely that the same surface and adsorbate will be in both the training and test sets (but not the exact same pairwise combination). However, if the approach must generalize to unseen surfaces the performance is much worse (OC20 results).

So is an approach that must have training data containing the surface in question useful? I would argue this would severely limit its practical utility. As an example, for the Alloy dataset there are ~6 samples per surface, which would likely result in 5 going to train and 1 going to test. So to get 66% accuracy you need 5 GMAE's on average for a surface in the training data - not a good trade-off. The paper would be much more convincing if the 8:1:1 split was done to hold out testing surfaces for FG and Alloy (i.e., the 8:1:1 was done on surfaces, not randomly per sample). This would demonstrate whether the approach generalizes to new surfaces outside of training.

The Uni-Mol+ help in providing a suitable baseline for the paper. Thank you for these results! The OC20 results should be included in the main text since this would be a likely "naive" approach for tackling the problem (the results on FG and Alloy don't make as much sense since they weren't trained on that data). It demonstrates that the problem is hard, and even these sampling approaches only get ~32% SR.

(Remarks on code availability)

.

Version 2:

Reviewer comments:

Reviewer #3

(Remarks to the Author)

I thank the reviewers for the additional results supplied in the latest revision. While the success rate decreased with the new splits, they are still reasonable. I would recommend replacing the results in the main manuscript with those using the surface type splits since this offers a more practically useful metric that others may follow.

The authors have addressed my concerns.

(Remarks on code availability)

Response to Reviews and List of Changes of the Manuscript NCOMMS-24-51449-T

Manuscript Number: NCOMMS-24-51449-T

Title: A multi-modal transformer for predicting global minimum adsorption energy

Author(s): Junwu Chen, Xu Huang, Cheng Hua, Yulian He and Philippe Schwaller

First of all, we thank the reviewers for their helpful comments. All Reviewers' comments have been taken into account point-by-point. Our detailed response to the Reviewers' comments and the list of changes (marked with highlighting in therevised manuscript) are provided below.

Reviewer #1

The paper presents a machine learning approach, AdsMT, for predicting the global minimum adsorption energy (GMAE) of catalyst surfaces. The proposed method uses a multi-modal transformer architecture that integrates surface graphs and adsorbate feature vectors through a cross-attention mechanism. This is a significant contribution. AdsMT predicts unseen surface-adsorbate combinations without binding site information. The method can also provide candidate adsorption sites based on the analysis of the attention weights. The integration of uncertainty quantification is also nice to experimentalists, as it provides a means of assessing the reliability of the model's predictions.

The flexibility of AdsMT is also noticeable. The three-part structure—graph encoder, vector encoder, and cross-modal encoder allows the choice of employing different graph encoders like SchNet, DimeNet++, and GemNet.

Despite the need of further performance enhancement for being useful for experimentalists, the paper is a significant contribution and would be of interest to readers of Nat Commun

1. While the AdsMT framework's performance on datasets like FG-GMAE and Alloy-GMAE is impressive, the results for the OCD-GMAE dataset are less promising. The authors mention that transfer learning improves performance on smaller datasets, but this aspect could be explored more deeply. It would be beneficial to have a more detailed analysis of why the transfer learning strategy leads to only moderate improvements in some cases and how the model could be further refined to address these challenges.

Response: We thank the reviewer for this comment. Our transfer learning obviously improved the performance of the AdsMT models on the OCD-GMAE dataset, but it only provides slight improvements on the FG-GMAE and Alloy-GMAE. The effectiveness of transfer learning mainly depends on the quality of pre-training data and the similarity between the source and target domains.

Regarding the quality of pre-training data, it is important to note the inherent difference between local minimum adsorption energy (LMAE) and global minimum adsorption energy (GMAE). We randomly sampled 100 surface-adsorbate combinations from OC20-LMAE and calculated their GMAE using the DFT method and AdsorbML pipeline [1]. The mean absolute difference between GMAE and LMAE is 0.27 eV. The MAEs of the AdsMT models without transfer learning on the OCD-GMAE dataset are about 0.6 eV. Therefore, it is reasonable that pre-training on LMAE data can improve the AdsMT performance on OCD-GMAE, despite the inherent difference between LMAE and GMAE. However, AdsMT already exhibits good predictive performance on the FG-GMAE and Alloy-GMAE datasets, with MAE values below or proximal to 0.1 eV. The inherent errors (> 0.2 eV) in LMAE data could hinder improving the AdsMT performance on both datasets through transfer learning.

The similarity between the source and target domains is also important for successful transfer learning. Both OCD-GMAE and OC20-LMAE datasets originate from the Open Catalyst Project with analogous surface and adsorbate types and the same DFT methods, which will be advantageous for transfer learning. The surfaces in both datasets were generated from crystals in the Materials Project database, which includes 54 elements, various crystal systems and space groups, and crystal facets with a maximum Miller index of less than 2. However, the surfaces in Alloy-GMAE were generated from binary alloys with face-centered cubic structures (Figure 1.1.1) and only used the (111) Miller index. The FG-GMAE dataset contains only 14 different surfaces from pure metals with a face-centered cubic structure. The significant differences in the distributions of surface compositions and geometries between the pre-training dataset and the target GMAE dataset could limit the effectiveness of transfer learning in enhancing AdsMT performance.

Figure 1.1.1: Crystal system distribution of bulk structures of surfaces in datasets.

Furthermore, we explore using universal machine learning interatomic potentials (MLIP) to obtain coarse GMAE data for model pre-training, which is much cheaper and potentially more accurate than the LMAE data. 10,000 surface-adsorbate combinations were sampled from OC20 and pretrained MLIP (EquiformerV2-31M) was used to compute coarse GMAE data through AdsorbML pipeline (num site=20) [1]. These data are named MLIP-GMAE and are used as resources for further pre-training. AdsMT models can be pre-trained on OC20-LMAE (~400k) first and then on MLIP-GMAE. As shown in Table 1.1.1-2, after further pre-training on the MLIP-GMAE, AdsMT performs better on the OCD-GMAE. Therefore, the coarse GMAE data from MLIP is also a valuable pre-training resource, which can further improve the performance of AdsMT on small datasets. Although this method is not cheap (about 80 GPU days for 10k data), it remains a promising direction for future exploration.

The detailed explanation on effective transfer learning was added to Section 2.3 of the manuscript and Supplementary Note 9.

Table 1.1.1: Test MAE results (eV) of AdsMT models with different graph encoders and pre-training conditions on the OCD-GMAE dataset.

Pretraining	AdsMT (GemNet-OC)	AdsMT (ET)	AdsMT (eSCN)	AdsMT (AdsGT)
None	0.645 ± 0.048	0.686 ± 0.065	0.589 ± 0.046	0.571 ± 0.057
OC20-LMAE	0.389 ± 0.048	0.395 ± 0.043	0.406 ± 0.037	0.430 ± 0.052
OC20-LMAE & MLIP-GMAE	0.352 ± 0.041	0.377 ± 0.038	0.382 ± 0.034	0.385 ± 0.046

Table 1.1.2: Test success rate results (%) of AdsMT models with different graph encoders and pre-training conditions on the OCD-GMAE dataset.

Pretraining	AdsMT (GemNet-OC)	AdsMT (ET)	AdsMT (eSCN)	AdsMT (AdsGT)
None	12.5 ± 2.3	11.9 ± 2.9	12.8 ± 2.0	13.5 ± 4.4
OC20-LMAE	22.0 ± 4.7	21.2 ± 3.3	21.0 ± 4.2	21.4 ± 3.6
OC20-LMAE & MLIP-GMAE	24.2 ± 3.3	23.8 ± 2.7	22.8 ± 2.9	23.1 ± 3.3

[1] Lan, J., Palizhati, A., Shuaibi, M. et al. AdsorbML: a leap in efficiency for adsorption energy calculations using generalizable machine learning potentials. *npj Comput. Mater.*, 2023, 9, 172. <https://doi.org/10.1038/s41524-023-01121-5>

2. The authors could also explore incorporating other forms of domain knowledge, such as geometric constraints or known adsorption site preferences, to improve the model's interpretability and accuracy.

Response: We thank the reviewer for this comment. We agree that highly relevant domain knowledge could improve the model's accuracy and interpretability. However, some chemical knowledge or descriptors, such as d-band centers and adsorption site preferences, are expensive, difficult to obtain, or not universal, limiting the model's practical application in virtual catalyst screening. Thus, we explored the introduction of domain knowledge to improve model performance through 1) 3D geometric descriptors of adsorbates, and 2) surface descriptors including site information.

1) The Mordred software (version 1.2.1) [1] and DFT-optimized geometries were used to calculate 3D geometric descriptors of the adsorbates. A total of 213 types of 3D descriptors were obtained, and the full descriptor list can be found in Mordred's documentation (<https://mordred-descriptor.github.io/documentation/master/descriptors.html>). The new 3D descriptors were merged into the original 2D adsorbate descriptors, and the performance of the AdsMT models was tested on the GMAE datasets. As shown in Tables 1.2.1-3, the additional 3D adsorbate descriptors can slightly improve AdsMT models' accuracy on OCD-GMAE and FG-GMAE. However, as shown in Figure 1.2.1, most of the adsorbates in the Alloy-GMAE dataset have only one atom and cannot produce useful 3D descriptors, which makes the additional 3D adsorbate descriptors ineffective for AdsMT on the Alloy-GMAE.

Table 1.2.1: Test MAE (eV) and success rate (SR, %) results of AdsMT models using different graph encoders on the OCD-GMAE dataset with and without 3D adsorbate descriptors.

	AdsMT (GemNet-OC)	AdsMT (ET)	AdsMT (eSCN)	AdsMT (AdsGT)
MAE w/o 3D desc.	0.645 ± 0.048	0.686 ± 0.065	0.589 ± 0.046	0.571 ± 0.057
MAE with 3D desc.	0.638 ± 0.047	0.682 ± 0.063	0.581 ± 0.043	0.564 ± 0.052
SR w/o 3D desc.	12.5 ± 2.3	11.9 ± 2.9	12.8 ± 2.0	13.5 ± 4.4
SR with 3D desc.	12.7 ± 2.3	11.9 ± 2.8	12.9 ± 1.8	13.7 ± 4.5

Table 1.2.2: Test MAE (eV) and success rate (SR, %) results of AdsMT models using different graph encoders on the Alloy-GMAE dataset with and without 3D adsorbate descriptors.

	AdsMT (GemNet-OC)	AdsMT (ET)	AdsMT (eSCN)	AdsMT (AdsGT)
MAE w/o 3D desc.	0.146 ± 0.010	0.146 ± 0.009	0.163 ± 0.011	0.143 ± 0.008
MAE with 3D desc.	0.149 ± 0.012	0.147 ± 0.010	0.161 ± 0.014	0.142 ± 0.011
SR w/o 3D desc.	65.0 ± 1.6	64.3 ± 1.7	56.8 ± 2.0	66.3 ± 1.3
SR with 3D desc.	64.8 ± 1.8	64.2 ± 1.8	56.6 ± 2.3	66.0 ± 1.2

Table 1.2.3: Test MAE (eV) and success rate (SR, %) results of AdsMT models using different graph encoders on the FG-GMAE dataset with and without 3D adsorbate descriptors.

	AdsMT (GemNet-OC)	AdsMT (ET)	AdsMT (eSCN)	AdsMT (AdsGT)
MAE w/o 3D desc.	0.097 ± 0.007	0.098 ± 0.008	0.102 ± 0.009	0.095 ± 0.007
MAE with 3D desc.	0.095 ± 0.006	0.097 ± 0.009	0.099 ± 0.008	0.092 ± 0.009
SR w/o 3D desc.	71.2 ± 2.3	71.6 ± 2.6	69.8 ± 2.9	71.9 ± 2.4
SR with 3D desc.	71.1 ± 2.4	71.7 ± 2.5	69.9 ± 2.6	72.1 ± 2.6

Figure 1.2.1: Distribution of atom numbers of adsorbates in GMAE datasets.

2) As shown in Table 1.2.4, surface descriptors containing chemical composition, geometry, and site information were explored to improve AdsMT performance. The additional surface descriptors were concatenated with adsorbate descriptors, and the performance of the AdsMT models was tested on the GMAE datasets. As shown in Tables 1.2.5-7, the additional surface descriptors had little or even negative effect on the accuracy of the AdsMT models. It suggests that the graph encoder in AdsMT can effectively capture information on composition, geometry, and local environment from the surface graphs. The additional surface descriptors could have minor or even negative effect on the performance of AdsMT models

Table 1.2.4: List of surface descriptors

Category	Descriptors	Software
chemical compositions	ElementProperty, Meredig, TMetalFraction, Stoichiometry, ElectronAffinity, ElectronegativityDiff, CohesiveEnergy, SiteElementalProperty	matminer [2]
surface & site geometries	SOAP, EwaldSiteEnergy, LocalPropertyDifference, OPSiteFingerprint	dscribe [3], matminer [2]

Table 1.2.5: Test MAE (eV) and success rate (SR, %) results of AdsMT models using different graph encoders on the Alloy-GMAE dataset with and without additional surface descriptors.

	AdsMT (GemNet-OC)	AdsMT (ET)	AdsMT (eSCN)	AdsMT (AdsGT)
MAE w/o surf. desc.	0.146 ± 0.010	0.146 ± 0.009	0.163 ± 0.011	0.143 ± 0.008
MAE with surf. desc.	0.145 ± 0.011	0.148 ± 0.011	0.160 ± 0.012	0.144 ± 0.009
SR w/o surf. desc.	65.0 ± 1.6	64.3 ± 1.7	56.8 ± 2.0	66.3 ± 1.3
SR with surf. desc.	65.1 ± 1.7	64.1 ± 1.6	56.7 ± 2.1	66.2 ± 1.1

Table 1.2.6: Test MAE (eV) and success rate (SR, %) results of AdsMT models using different graph encoders on the FG-GMAE dataset with and without additional surface descriptors.

	AdsMT (GemNet-OC)	AdsMT (ET)	AdsMT (eSCN)	AdsMT (AdsGT)
MAE w/o surf. desc.	0.097 ± 0.007	0.098 ± 0.008	0.102 ± 0.009	0.095 ± 0.007
MAE with surf. desc.	0.096 ± 0.007	0.099 ± 0.010	0.105 ± 0.007	0.096 ± 0.007
SR w/o surf. desc.	71.2 ± 2.3	71.6 ± 2.6	69.8 ± 2.9	71.9 ± 2.4
SR with surf. desc.	71.5 ± 2.2	71.4 ± 2.4	69.6 ± 2.8	71.7 ± 2.5

Table 1.2.7: Test MAE (eV) and success rate (SR, %) results of AdsMT models using different graph encoders on the OCD-GMAE dataset with and without additional surface descriptors.

	AdsMT (GemNet-OC)	AdsMT (ET)	AdsMT (eSCN)	AdsMT (AdsGT)
MAE w/o surf. desc.	0.645 ± 0.048	0.686 ± 0.065	0.589 ± 0.046	0.571 ± 0.057

MAE with surf. desc.	0.648 ± 0.051	0.681 ± 0.062	0.593 ± 0.045	0.568 ± 0.058
SR w/o surf. desc.	12.5 ± 2.3	11.9 ± 2.9	12.8 ± 2.0	13.5 ± 4.4
SR with surf. desc.	12.2 ± 2.4	12.1 ± 3.0	12.6 ± 2.2	13.4 ± 4.6

[1] Moriwaki, H., Tian, YS., Kawashita, N. et al. Mordred: a molecular descriptor calculator. *J. Cheminform.*, 2018, 10, 4. <https://doi.org/10.1186/s13321-018-0258-y>

[2] Ward, L., Dunn, A., Faghaninia, A., Zimmermann, . et al., Matminer: An open source toolkit for materials data mining. *Comput. Mater. Sci.*, 2018, 152, 60-69. <https://doi.org/10.1016/j.commatsci.2018.05.018>

[3] Himanen, Lauri, et al., Dscribe: Library of descriptors for machine learning in materials science. *Computer Physics Communications*, 2020, 247, 106949. <https://doi.org/10.1016/j.cpc.2019.106949>

Reviewer #2

Chen et al. developed a transformer model to compute the global minimum adsorption energy of small adsorbates on metal (and metal alloy) surfaces. The model uses surface information (in the form of graphs), adsorbate information (using molecular descriptors), and multi-modal learning to compute the global minimum energy value for an adsorbate-surface pair. The resulting model offers some predictive capabilities in and of itself but with transfer learning from local minimum energy structures is able to have prediction errors between 0.1 – 0.4 eV. While these errors are still on the higher side, this model is quite useful from the standpoint of quick screening knowing only the surface information and basic adsorbate descriptors and not needing detailed adsorption structure knowledge. Indeed, the authors show that even using machine-learned potentials, computing global minimum adsorption structures can be time-consuming compared to using their model. For this reason, this work is interesting and worthy of publication in this journal, however, after consideration of the following questions and clarifications.

1. The authors choose multi-modal transformer learning and a specific type of self- and cross-attention model structure. It is unclear why this technique is even necessary considering the limited size of the data set. Are there comparisons of model performance against some benchmarks (similar techniques that only predict global minimum energy structures? Or comparison with ML potentials, although it's not a perfect one-to-one comparison.

Response: We thank the reviewer for this comment. Our manuscript aims to directly predict global minimum adsorption energy (GMAE) using separate surface and adsorbate information without requiring site-binding information, making it suitable for efficient virtual screening of catalysts where adsorption structures are rarely available. Therefore, our goal requires models with strong learning capabilities to capture complex interaction information between two objects (surface and adsorbate). According to Lee et al. [1], the combination of self-attention and cross-attention layers enables excellent learning of complex associations between two modalities/objects and precise target prediction, significantly outperforming multilayer perceptrons (MLP). To explore the performance of self & cross-attention on GMAE prediction, we compared it with MLP that utilizes surface graph embeddings and adsorbate descriptors to predict GMAE. As shown in Table 2.1.1-2.1.3, the self & cross-attention obviously outperform the MLP on GMAE predictions, even with a similar number of parameters. Moreover, the cross-attention layers use surface atom embeddings and adsorbate features as input, and can output attention weights mapping from each surface atom to adsorbate. This endows our AdsMT model with the interpretable potential to infer optimal adsorption sites. In summary, the self- and cross-attention layers are of great significance to provide both high prediction accuracy and interpretability presented in this work.

Table 2.1.1: Test MAE results (eV) of AdsMT models with SchNet graph encoder using MLP or self&cross-attention layers to predict GMAE.

	OCD-GMAE	FG-GMAE	Alloy-GMAE
MLP	0.895 ± 0.081	0.183 ± 0.021	0.232 ± 0.034
self&cross-attention	0.661 ± 0.056	0.103 ± 0.012	0.162 ± 0.010

Note: i) 4-layer MLP (336, 512, 512, 512, 1), param: 701k

ii) 1-layer self-attention & 1-layer cross-attention (heads=4), param: 329k

Table 2.1.2: Test MAE results (eV) of AdsMT models with GemNet-OC graph encoder using MLP or self&cross-attention layers to predict GMAE.

	OCD-GMAE	FG-GMAE	Alloy-GMAE
MLP	0.976 ± 0.064	0.168 ± 0.018	0.217 ± 0.027
self&cross-attention	0.645 ± 0.048	0.097 ± 0.007	0.146 ± 0.010

Note: i) 4-layer MLP (336, 512, 512, 512, 1), param: 701k

ii) 1-layer self-attention & 1-layer cross-attention (heads=4), param: 329k

Table 2.1.3: Test MAE results (eV) of AdsMT models with AdsGT graph encoder using MLP or self&cross-attention layers to predict GMAE.

	OCD-GMAE	FG-GMAE	Alloy-GMAE
MLP	0.787 ± 0.069	0.146 ± 0.032	0.208 ± 0.041
self&cross-attention	0.571 ± 0.057	0.095 ± 0.007	0.143 ± 0.008

Note: i) 4-layer MLP (336, 512, 512, 512, 1), param: 701k

ii) 1-layer self-attention & 1-layer cross-attention (heads=4), param: 329k

[1] Lee, Namkyeong, et al., Density of states prediction of crystalline materials via prompt-guided multi-modal transformer, 37th Conference on Neural Information Processing Systems (NeurIPS 2023) <https://openreview.net/forum?id=2lWh1G1W1l>

2. The authors correctly do not use structural information of adsorbates while building the adsorbate encoder but refer to RDKit-derived features. They should explain further what these features are and whether they need 3D structure information (and if yes, how do they get this). In a real-world application, the user of such a model may only hypothesize the species connectivity (e.g. they may be able to say the relevant intermediates are NH₂ or CHCH₂) but not be able to provide specific structures.

Response: We thank the reviewer for this comment. Our model utilizes 2D RDKit-derived features without requiring the geometric structure of the adsorbate. All adsorbate features used in AdsMT were listed in Table 2.2.1, which were also added to the Supplementary Information Table S5.

Table 2.2.1: List of adsorbate descriptors used in the AdsMT model.

Descriptor/Descriptor Family	Notes
Gasteiger/Marsili Partial Charges	Tetrahedron 36 :3219 – 28 (1980)
BalabanJ	Chem. Phys. Lett. 89 :399 – 404 (1982)

BertzCT	J. Am. Chem. Soc. 103 :3599 – 601 (1981)
Ipc	J. Chem. Phys. 67 :4517 – 33 (1977)
HallKierAlpha	Rev. Comput. Chem. 2 :367 – 422 (1991)
Kappa1 – Kappa3	Rev. Comput. Chem. 2 :367 – 422 (1991)
Phi	New in 2021.03 release Quant. Struct.– Act. Rel. 8 :221 – 224 (1989)
Chi0, Chi1	Rev. Comput. Chem. 2 :367 – 422 (1991)
Chi0n – Chi4n	Rev. Comput. Chem. 2 :367 – 422 (1991)
Chi0v – Chi4v	Rev. Comput. Chem. 2 :367 – 422 (1991)
MolLogP	Wildman and Crippen JCICS 39 :868 – 73 (1999)
MolMR	Wildman and Crippen JCICS 39 :868 – 73 (1999)
MolWt	Average molecular weight of the molecule
ExactMolWt	Exact molecular weight of the molecule
HeavyAtomCount	Number of heavy atoms in a molecule
HeavyAtomMolWt	Average molecular weight of the molecule ignoring hydrogens
NHOHCount	Number of NHs or OHs
NOCCount	Number of nitrogens and oxygens
NumHAcceptors	Number of hydrogen bond acceptors
NumHDonors	Number of hydrogen bond donors
NumHeteroatoms	Number of heteroatoms
NumRotatableBonds	Number of rotatable bonds
NumValenceElectrons	Number of valence electrons the molecule has
NumAmideBonds	Number of amide bonds
Num{Aromatic,Saturated,Aliphatic}Rings	Number of aromatic/ saturated/ aliphatic(at least one non-aromatic bond) rings in a molecule
Num{Aromatic,Saturated,Aliphatic}{Hetero,Carbo}cycles	Number of aromatic/ saturated/ aliphatic(at least one non-aromatic bond) heterocycles/ carbocycles in a molecule
RingCount	Number of rings in a molecule
FractionCSP3	Fraction of carbons that are sp ³ hybridized
NumSpiroAtoms	Number of spiro atoms (atoms shared between rings that share one atom)
NumBridgeheadAtoms	Number of bridgehead atoms (atoms shared between rings that share at least two bonds)
TPSA	J. Med. Chem. 43 :3714 – 7, (2000) See the section in the RDKit book describing differences to the original publication.

LabuteASA	J. Mol. Graph. Mod. 18 :464 – 77 (2000)
PEOE_VSA1 – PEOE_VSA14	MOE-type descriptors using partial charges and surface area contributions http://www.chemcomp.com/journal/vsadesc.htm
SMR_VSA1 – SMR_VSA10	MOE-type descriptors using MR contributions and surface area contributions http://www.chemcomp.com/journal/vsade-sc.htm
SlogP_VSA1 – SlogP_VSA12	MOE-type descriptors using LogP contributions and surface area contributions http://www.chemcomp.com/journal/vsade-sc.htm
EState_VSA1 – EState_VSA11	MOE-type descriptors using EState indices and surface area contributions (developed at RD, not described in the CCG paper)
VSA_EState1 – VSA_EState10	MOE-type descriptors using EState indices and surface area contributions (developed at RD, not described in the CCG paper)
MQNs	Nguyen et al. ChemMedChem 4 :1803 – 5 (2009)
Topliss fragments	implemented using a set of SMARTS definitions in \$(RDBASE)/Data/FragmentDescriptors.csv
Autocorr2D	New in 2017.09 release. Todeschini and Consoni “Descriptors from Molecular Geometry” Handbook of Chemoinformatics https://doi.org/10.1002/9783527618279.ch37
BCUT2D	New in 2020.09 release. Pearlman and Smith in “3D-QSAR and Drug design: Recent Advances” (1997)

3. The surface graph seems to be generated from a periodic structure available from the OC database. Therefore, how sensitive are the predictions to the atom distances (or broadly speaking lattice constants)? A user may want to rely only on connectivity information (and approximate lattice constant information from a database such as ICSD, Materials project, AFLOWlib, etc.) for a new structure (not in the OC database). Ideally, therefore, the model should be insensitive (or only mildly sensitive) to atom-atom distances on the surface.

Response: We thank the reviewer for this comment. To study the sensitivity of the AdsMT model to the surface’s geometry (e.g., atom distances and lattice constants), we added noise to the input surface structures without changing atomic connectivity through the following methods:

- 1) randomly perturbing atomic coordinates in the x, y, and z directions (displacements < 0.2 Å);
- 2) randomly perturbing the lengths of lattice vectors *a* and *b* (deviations < 2%). The *c* vector, which corresponds to the vacuum layer direction, was unchanged.

By comparing GMAE predictions from the original and noise-perturbed surface structures, we evaluated the prediction fluctuations and mean energy differences induced by geometric variations. As shown in Table 2.3.1, AdsMT models with different graph encoders exhibit low energy prediction fluctuation after adding geometric noise to the input surfaces. In particular on the FG-GMAE dataset, the mean absolute energy differences of GMAE predictions between original and noise-perturbed surfaces are very small, only about 0.014 eV. The results show that our model is insensitive to geometric variations of input surfaces where atomic connectivity does not change. This insensitivity is advantageous for AdsMT models in virtual catalyst screening, where surface structures could be generated from different crystal databases. These explorations and results were added to the Discussion of our manuscript and Supplementary Note 11.

Table 2.3.1: Mean absolute energy differences (eV) of GMAE predictions between original and noise-perturbed surface structures using the same trained models. For each surface, 10 noise-perturbed surfaces were generated for evaluation of GMAE prediction fluctuations.

Graph encoders	Alloy-GMAE	FG-GMAE	OCD-GMAE
SchNet	0.053	0.016	0.077
CGCNN	0.047	0.011	0.068
DimNet++	0.068	0.021	0.073
GemNet-OC	0.054	0.016	0.046
ET	0.044	0.014	0.054
eSCN	0.028	0.009	0.045
AdsGT	0.031	0.012	0.052
Average	0.046	0.014	0.059

4. Using the attention scores to suggest which surface atoms are likely to participate in binding is interesting but given that the model never takes any information about adsorbate binding as input, this “interpretation” is probably just some kind of correlation that the model learns (e.g. whenever there are surface Cr atoms or N atoms, the binding energy may be high). In any case, this model is likely not able to say if the preferred adsorption mode is monodentate or bidentate, or if the site is a bridge, three-fold, or a top site, etc. In that sense, are the authors overinterpreting the capabilities of the model?

Response: We thank the reviewer for this comment. Recently, many studies on attention score-based interpretability have been reported across fields such as natural language processing, computer vision, and chemistry [1-4]. For example, Schwaller et al. [1] reported the interpretability of attention scores from Transformer models for identifying atom rearrangements in reactions using unsupervised schemes. Therefore, this manuscript investigated whether the cross-attention layers can learn important chemical information and apply attention scores to infer the optimal adsorption site. Our method can reveal the surface atom importances or the optimal surface atoms for stronger binding energy, which are highly related to but not strictly equivalent to the actual adsorption site. To ensure precise expression, we revised the term "interpretability" in the manuscript to "interpretable potential". Furthermore, we explored extending our method to identify different types of adsorption sites (e.g., top, bridge, hollow) rather than simply assessing the importance of individual surface atoms. Given a set of surface atoms \mathbf{A} and their attention scores \mathbf{S} , higher attention scores indicate that the atom is more important for strong adsorption. The maximum attention score is defined as s_{max} , and the set of atoms with scores above $0.8s_{max}$ is defined as the potential site atoms \mathbf{P} . For the atom i in \mathbf{P} , if no atom j in \mathbf{P} satisfies Equation (1), then atom i is considered to constitute a top site. For the atom i in \mathbf{P} , if there is only one atom j in \mathbf{P} that satisfies Equation (1), then atoms i and j are considered to constitute a bridge site. If there are three atoms i , j and k in \mathbf{P} , where d_{ij} , d_{ik} and d_{jk} all satisfy Equation (1) and form a triangle, then atoms i , j and k are considered to constitute a hollow site.

$$d_{ij} < r_i + r_j + \delta, \quad i \neq j \quad (1)$$

In Equation (1), d_{ij} represents the distance between atoms i and j , r_i and r_j are the covalent radii of atoms i and j , respectively, and δ can either be a fixed value or the geometric diameter of the adsorbate. We evaluated this extended method using our AdsMT models, without considering more complex site types (e.g., 4-fold hollow site). As shown in Figure 2.4.1, based on the improved method described above, it is possible to identify adsorption site types (top, bridge, hollow) using

attention scores and the distances between high-scoring atoms. Isolated high-scoring atoms without surrounding high-scoring atoms tend to form top sites, while high-scoring atoms close to each other are more likely to form bridge or hollow sites. The results show that the cross-attention scores have the potential to identify the adsorption site type through the improved method, which will be further studied in future work. The description of this extension method has been added to Supplementary Note 10 and Figure S14.

Figure 2.4.1: Examples of identifying different adsorption site types based on the extended method and cross-attention scores. Left: cross-attention score-colored surfaces. Right: DFT-optimized adsorption structures at GMAE.

[1] Philippe Schwaller et al., Extraction of organic chemistry grammar from unsupervised learning of chemical reactions. *Sci. Adv.*, 2021, 7, eabe4166. DOI: <https://doi.org/10.1126/sciadv.abe4166>

[2] Mattia Rigotti, et al., Attention-based Interpretability with Concept Transformers, International Conference on Learning Representations (ICLR 2022) <https://openreview.net/forum?id=kAa9eDS0RdO>

[3] Chefer, H., et al., Transformer interpretability beyond attention visualization. Proceedings of the IEEE/CVF Conference on Computer Vision and Pattern Recognition (CVPR), 2021, pp. 782-791. <https://doi.org/10.1109/CVPR46437.2021.00084>

[4] S. Liu, et al., On Exploring Attention-based Explanation for Transformer Models in Text Classification, 2021 IEEE International Conference on Big Data (Big Data), 2021, pp. 1193-1203, <https://doi.org/10.1109/BigData52589.2021.9671639>

5. It was interesting to see that the model was able to learn from the local minimum structures. The naïve thinking would have been that such transfer learning would actually be detrimental because the LMAE dataset would have a lot of a one-to-many mapping (one adsorbate-surface

combination having multiple energy values). Any thoughts on why this transfer learning works? Or does it work only because the encoding part of the model is learned using this pre-training step? Can the authors also clarify if there are adsorbate-surface combinations in the LMAE dataset that are not in the GMAE dataset?

Response: We thank the reviewer for this comment. To prevent harmful pre-training, we performed strict data cleaning on the OC20 IS2RE dataset to obtain the OC20-LMAE dataset (more details in Methods section):

1) The data points with anomalies (adsorbate desorption/dissociation, surface reconstruction) are removed.

2) If a surface-adsorbate combination has multiple local adsorption energies, the minimum value is selected as the prediction target. Therefore, in the OC20-LMAE dataset, each surface-adsorbate combination has only one energy value.

3) If a surface-adsorbate combination also exists in the GMAE datasets, this combination is removed from the OC20-LMAE dataset. Thus, all surface-adsorbate combinations in the OC20-LMAE dataset do not exist in the GMAE datasets.

After freezing the parameters of some graph convolutional layers of the graph encoder, the pre-trained AdsMT model is fine-tuned on the GMAE datasets. The model performance improvement from transfer learning could be attributed to the following two reasons:

1) pre-training on a large number of surface-adsorbate combinations can help the graph encoder learn to capture important chemical information from the surface structure for adsorption energy prediction;

2) pre-training can help the cross-modal encoder learn to predict GMAE from the embeddings of the surface graphs and the adsorbate features.

Reviewer #3

The paper addresses an important problem in catalysis of finding the global minimum adsorption energy for an adsorbate on a catalyst's surface. They propose a technique that first computes a graph embedding for the catalyst and a vector embedding for the adsorbate. These are then combined with attention models to predict the energy.

Results are demonstrated on 3 different datasets with and without transfer learning. The results on dataset with lower diversity are reasonable with a success rate of 66-72%. However, success on the OCD dataset which has a much larger variety of catalysts was only 14-17%.

The paper has two main issues that limit its potential to be published:

1) Since the success rates are low and the relaxed structures are not provided by the approach, how would the approach be used in practice? If relaxed structures are provided, DFT can be performed to double check their correctness. Without them, a full DFT relaxation would need to be performed to check the final results (assuming the adsorbate site is correctly found, otherwise it's even worse). Practically speaking if a DFT relaxation take 200 steps, an approach that produces relaxed structures gets 200 more "tries" than an approach that doesn't.

Response: We thank the reviewer for this comment. At the current stage, our AdsMT showed good accuracy and fast calculation speed (SI Note 7) in predicting GMAE on medium-sized Alloy-GMAE and FG-GMAE datasets without need of site-binding information. Such a model would allow rapid screening of promising catalysts, especially alloy-based catalysts, from existing materials databases (e.g., ICSD, Materials Project). Fast prediction of adsorption energy is particularly important for the virtual screening of catalysts, while the relaxed adsorption structure is more important for the study of catalytic mechanisms. Based on the Sabatier principle (Figure 3.1.1), the highest catalytic activity is usually achieved when the adsorption energy of the key intermediate is at the optimal value. Therefore, in this work and many others [1-6], the primary interest was positioned to predict the adsorption energy in a fast and accurate manner, without

providing relaxed adsorption structures. We acknowledge that the AdsMT’s performance on the OCD-GMAE is suboptimal and needs to be improved, mainly due to the small dataset size with only 973 combinations but a wider chemical distribution. This is commonly seen in many ML models applied on small datasets. We also attempted to improve its accuracy using transfer learning, ending up with a higher success rate of 22%. The performance can be further enhanced by filtering out unreliable predictions and retaining predictions with low uncertainty through model ensemble and uncertainty estimation. As shown in Figure 3.1.2, AdsMT’s predictions with lower uncertainty tend to have a higher success rate across three GMAE datasets. When taking the 10% predictions with the lowest uncertainty, AdsMT can achieve about 80% and 90% success rates on the Alloy-GMAE and FG-GMAE, respectively. Moreover, AdsMT can reach about 50% success rate on the OCD-GMAE when only retaining 10% predictions with the lowest uncertainty.

Figure 3.1.1: Schematic representation of the Sabatier principle.

Figure 3.1.2: Cumulative success rate (%) at different cutoffs of uncertainty percentiles, in uncertainty estimation of AdsMT ensembles (n=10) with different graph encoders.

Combined with the low prediction cost, AdsMT model can be used for rapid preliminary screening in large catalyst libraries and pinpoint a small range of candidate catalyst surfaces with low uncertainty and desired GMAE of key reaction intermediates. Afterwards, more precise methods such as DFT can be used to further validate the top candidate catalysts. For example, given a target reaction involving two key intermediate adsorbates and a database containing 200,000 catalysts, assuming 20 surface structures are generated from each catalyst, a total of 4,000,000 possible surfaces can be obtained. For the AdsMT model, it would only require around 1 GPU day to compute the GMAE of all possible surfaces for the two key adsorbates. However, it would spend about 37037 GPU days using pre-trained machine learning interatomic potentials (GemNet-OC-39M) with AdsorbML pipeline [7]. Even using 100 GPUs simultaneously, it still needs ~370 days to complete all calculations. Therefore, the AdsMT model is more cost-effective and holds greater potential for large-scale catalyst pre-screening, especially when dealing with large

catalyst databases ($> 1\text{M}$). Finally, after uncertainty estimation and removal of unreliable predictions, the DFT method can be used to further validate the top 100 candidate catalysts.

Moreover, regardless of whether the model outputs relaxed adsorption structures, the DFT ground truth of GMAE must be obtained by enumerating initial configurations and DFT relaxations (Path 1 in Figure 3.1.3). Even if the model provides adsorption structures, there is no guarantee that these structures are close to true global or local minima, and the ground truth of GMAE cannot be obtained from one DFT single-point or relaxation based on the provided structures. Therefore, both models with and without providing relaxed structures require the same DFT workflow and computation time to obtain the GMAE ground truth and evaluate the predicted GMAE.

Figure 3.1.3: Schematic diagram of different routes to obtain ground truth or ML-based global minimum adsorption energy (GMAE).

- [1] Zhong, M., Tran, K., Min, Y. et al. Accelerated discovery of CO₂ electrocatalysts using active machine learning. *Nature*, 2020 581, 178–183. <https://doi.org/10.1038/s41586-020-2242-8>
- [2] Christopher C. Price et al., Efficient catalyst screening using graph neural networks to predict strain effects on adsorption energy. *Sci. Adv.*, 2022, 8, eabq5944. <https://doi.org/10.1126/sciadv.abq5944>
- [3] Mok, D.H., Li, H., Zhang, G. et al. Data-driven discovery of electrocatalysts for CO₂ reduction using active motifs-based machine learning. *Nat. Commun.*, 2023, 14, 7303. <https://doi.org/10.1038/s41467-023-43118-0>
- [4] Fung, V., Hu, G., Ganesh, P. et al. Machine learned features from density of states for accurate adsorption energy prediction. *Nat. Commun.*, 2021, 12, 88. <https://doi.org/10.1038/s41467-020-20342-6>
- [5] Machine Learning-Driven High-Throughput Screening of Alloy-Based Catalysts for Selective CO₂ Hydrogenation to Methanol, *ACS Appl. Mater. Interfaces*, 2021, 13, 47, 56151–56163. <https://doi.org/10.1021/acsami.1c16696>
- [6] Pablo-García, S., Morandi, S., Vargas-Hernández, R.A. et al. Fast evaluation of the adsorption energy of organic molecules on metals via graph neural networks. *Nat. Comput. Sci.*, 2023, 3, 433–442. <https://doi.org/10.1038/s43588-023-00437-y>
- [7] Lan, J., Palizhati, A., Shuaibi, M. et al. AdsorbML: a leap in efficiency for adsorption energy calculations using generalizable machine learning potentials. *npj Comput. Mater.*, 2023, 9, 172. <https://doi.org/10.1038/s41524-023-01121-5>

2) Since this paper emphasizes performance gains, other baselines should be considered. For instance, IS2RE models that directly predict relaxed energies from initial structures (no relaxation is performed) could be used along with sampling 2-10 random adsorbate positions. This would be

marginally slower than the authors approach and may be more accurate. The reason for this is there exists a lot more data for training an IS2RE model. Training a model using the OC20 Dense dataset is very challenging due to its small size, which was originally intended as a testing dataset, not a training dataset.

Response: We thank the reviewer for this comment. To compare with the AdsMT model (~5M) on the GMAE task, we selected the Uni-Mol+ model (49M) [1, 2], which is one of the best models for directly predicting local relaxed energies from initial structures in the OC20 IS2RE task. The test energy MAE of the Uni-Mol+ model is 0.4143 eV on the OC20 IS2RE task. [1, 2] As shown in Table 3.2.1–3.2.3, the success rate of pretrained Uni-Mol+ increases as the number of random initial structures grows. When the number of initial structures exceeds 50, the Uni-Mol+ model outperforms the AdsMT model on the OCD-GMAE. In contrast, the calculation speed of the AdsMT model is much faster than the Uni-Mol+ model. For Alloy-GMAE and FG-GMAE, the Uni-Mol+ model exhibits low success rates, even with 100 initial structures for each surface-adsorbate combination. This may be attributed to large differences between the OC20 dataset and these two datasets (such as DFT methods, surface types, and adsorbate types). In addition, many catalyst datasets do not provide DFT calculation trajectories, making it challenging to fine-tune Uni-Mol+ or machine learning potential models on these datasets. Although the AdsMT’s accuracy on OCD-GMAE needs to be improved, it achieves good performance on Alloy-GMAE and FG-GMAE datasets. Among them, alloy is one of the most commonly used catalysts with a large number of possible components and structures. Combined with uncertainty estimation, AdsMT is promising and useful for the virtual screening of alloy catalysts.

Table 3.2.1: Comparison of the success rate (%) and computational speed (comb/min) of AdsMT and pretrained Uni-Mol+ on the OCD-GMAE dataset.

Models	Uni-Mol+				AdsMT after TL
Initial structures	10	25	50	100	—
Success rate (%)	14.8	20.6	29.8	32.7	22.0
Speed (comb./min)	107.1	42.8	21.4	10.7	4902.4

Table 3.2.2: Comparison of the success rate (%) and computational speed (comb/min) of AdsMT and pretrained Uni-Mol+ on the Alloy-GMAE dataset.

Models	Uni-Mol+				AdsMT after TL
Initial structures	10	25	50	100	—
Success rate (%)	7.6	12.8	15.7	16.1	68.4
Speed (comb./min)	107.1	42.8	21.4	10.7	4902.4

Table 3.2.3: Comparison of the success rate (%) and computational speed (comb/min) of AdsMT and pretrained Uni-Mol+ on the FG-GMAE dataset.

Models	Uni-Mol+				AdsMT after TL
Initial structures	10	25	50	100	—
Success rate (%)	6.8	8.3	9.4	11.6	72.6
Speed (comb./min)	107.1	42.8	21.4	10.7	4902.4

- [1] Lu, S., Gao, Z., He, D. et al. Data-driven quantum chemical property prediction leveraging 3D conformations with Uni-Mol+. Nat Commun 15, 7104 (2024). <https://doi.org/10.1038/s41467-024-51321-w>
- [2] Lu, Shuqi, et al. Highly accurate quantum chemical property prediction with Uni-Mol+, arXiv preprint arXiv:2303.16982 (2023). <https://arxiv.org/abs/2303.16982>

Minor comments:

1) Are catalyst surface and adsorbate features really multi modal? In ML when I hear multi modal I usually think of images vs. text or similar differences. They are both atom representations, and it is unclear how much advantage is gained from treating them separately. For instance the adsorbate could be encoded using the mean atom embedding?

Response: We thank the reviewer for this comment. Common multimodal models use modalities such as text, images, and audio. However, in a broad sense, multimodal models refer to models that can integrate and process multiple types of data, where the data types are not limited to human-perceivable types. For instance, Kim et al. [1] created a multi-modal pre-training transformer that integrates atom-wise graphs (graph-type data) and energy-grid embeddings (matrix-type data) to predict the properties of metal-organic frameworks (MOFs). Park et al. [2] proposed a multi-modal transformer with outstanding performance in predicting the density of states (DOS) through modalities of crystal graphs (graph-type data) and crystal orbital energies (matrix-type data). Matrices are Euclidean data, while graphs are non-Euclidean data [3]. Our AdsMT model uses surface graphs (graph-type data) and adsorbate descriptors (matrix-type data) to predict the global minimum adsorption energy (GMAE). Therefore, it is reasonable to refer to it as a multimodal model.

Some important adsorbates (e.g., *H, *O, *N) have only one atom, and message passing in GNNs cannot work for one node without edge. Many adsorbate species (e.g., *CO₂, *CO, *OH, *NH₂) consist of fewer than four atoms or two bonds, which makes capturing important chemical information difficult through atomic representation and graph learning. In addition, most catalyst datasets contain less than 100 types of adsorbates, and some datasets may have less than 10 types. Such a limited variety of adsorbates can also hinder graph learning. On the other hand, molecular descriptors based on expert knowledge can quickly and accurately capture the chemical information of adsorbates, especially for small adsorbates or new adsorbates without structural information. Therefore, we used molecular descriptors to represent adsorbates rather than graphs or atom representation. Moreover, utilizing separate information of surfaces and adsorbates as input, the AdsMT can predict GMAE without the need of binding information and is able to generalize to unseen surface-adsorbate combinations, making it suitable for efficient virtual screening of catalysts where adsorption structures are rarely available. These detailed reasons were also added to Methods 4.3 section of the manuscript.

- [1] Kang, Y., Park, H., Smit, B. et al., A multi-modal pre-training transformer for universal transfer learning in metal-organic frameworks., Nat. Mach. Intell., 2023, 5, 309-318. <https://doi.org/10.1038/s42256-023-00628-2>
- [2] Lee, Namkyeong, et al., Density of states prediction of crystalline materials via prompt-guided multi-modal transformer, 37th Conference on Neural Information Processing Systems (NeurIPS 2023) <https://openreview.net/forum?id=2lWh1G1W1l>
- [3] Z. Wu, S. Pan, et al., A Comprehensive Survey on Graph Neural Networks, IEEE Transactions on Neural Networks and Learning Systems, 2021, 32, 1, 4-24. <https://doi.org/10.1109/TNNLS.2020.2978386>

2) What are the eSCN results after transfer learning? From the bar graphs you can't determine whether it outperforms AdsGT.

Response: We thank the reviewer for this comment. As shown in Table 3.4.1-3, after transfer learning, the eSCN encoder surpasses the AdsGT encoder on the OCD-GMAE dataset, while the AdsGT encoder still outperforms the eSCN encoder on the Alloy-GMAE and FG-GMAE datasets. These results of AdsMT models with and without (w/o) transfer learning were listed in Supplementary Information Table S8-10.

Table 3.4.1: Performance comparison of MAE (unit: eV) and success rate (SR, unit: %) for AdsMT models adopting different graph encoders w/o and with transfer learning (TL) on the OCD-GMAE dataset.

	GemNet-OC	ET	eSCN	AdsGT
MAE w/o TL	0.645 ± 0.048	0.686 ± 0.065	0.589 ± 0.046	0.571 ± 0.057
MAE with TL	0.389 ± 0.048	0.395 ± 0.043	0.406 ± 0.037	0.430 ± 0.052
MAE decrease	0.256	0.291	0.183	0.141
SR w/o TL	12.5 ± 2.3	11.9 ± 2.9	12.8 ± 2.0	13.5 ± 4.4
SR with TL	22.0 ± 4.7	21.2 ± 3.3	21.0 ± 4.2	21.4 ± 3.6
SR increase	9.5	9.3	8.2	7.9

Table 3.4.2: Performance comparison of MAE (unit: eV) and success rate (SR, unit: %) for AdsMT models adopting different graph encoders w/o and with transfer learning (TL) on the Alloy-GMAE dataset.

	GemNet-OC	ET	eSCN	AdsGT
MAE w/o TL	0.146 ± 0.010	0.146 ± 0.009	0.163 ± 0.011	0.143 ± 0.008
MAE with TL	0.141 ± 0.011	0.142 ± 0.009	0.153 ± 0.006	0.140 ± 0.010
MAE decrease	0.005	0.004	0.01	0.003
SR w/o TL	65.0 ± 1.6	64.3 ± 1.7	56.8 ± 2.0	66.3 ± 1.3
SR with TL	66.5 ± 1.5	67.0 ± 1.1	60.8 ± 1.1	68.4 ± 0.8
SR increase	1.5	2.7	4	2.1

Table 3.4.3: Performance comparison of MAE (unit: eV) and success rate (SR, unit: %) for AdsMT models adopting different graph encoders w/o and with transfer learning (TL) on the FG-GMAE dataset.

	GemNet-OC	ET	eSCN	AdsGT
MAE w/o TL	0.097 ± 0.007	0.098 ± 0.008	0.102 ± 0.009	0.095 ± 0.007
MAE with TL	0.095 ± 0.008	0.097 ± 0.007	0.096 ± 0.007	0.094 ± 0.008
MAE decrease	0.002	0.001	0.006	0.001
SR w/o TL	71.2 ± 2.3	71.6 ± 2.6	69.8 ± 2.9	71.9 ± 2.4
SR with TL	72.3 ± 2.3	72.0 ± 2.1	72.2 ± 2.7	72.6 ± 1.8
SR increase	1.1	0.4	2.4	0.7

3) How was “The data cleaning was conducted to remove abnormal adsorption structures” performed for the datasets? How many data points remained? This makes it hard to compare against previous papers.

Response: We thank the reviewer for this comment. Based on AdsorbML [1], we adopt several constraints of the relaxed adsorbate-surface structures to obtain valid adsorption energies. The relaxed structures with adsorbate desorption/dissociation or surface mismatch are regarded as abnormal adsorption structures. Specifically, 1) adsorbate should not be desorbed from the surface in the final relaxed structure; 2) adsorbate should not dissociate or break apart into multiple adsorbates because it would no longer be the adsorption energy of the molecule of interest; and 3) if the adsorbate induces significant changes in the surface compared to the clean surface, the Eslab reference would create a surface mismatch.[1] However, during data cleaning, we found that the original datasets had already eliminated abnormal adsorption structures and we did not need to delete any local adsorption energy data. In fact, our data cleaning was conducted to sort the local adsorption energies and take the lowest adsorption energy of all relaxed structures as the global minimum adsorption energy (GMAE) of each adsorbate-surface combination. This makes our results easily comparable to previous papers. Moreover, we revised the description of data cleaning in the Methods section of our manuscript.

[1] Lan, J., Palizhati, A., Shuaibi, M. et al. AdsorbML: a leap in efficiency for adsorption energy calculations using generalizable machine learning potentials. *npj Comput. Mater.*, 2023, 9, 172. <https://doi.org/10.1038/s41524-023-01121-5>

4) The authors state "To the best of our knowledge, this is the first work that directly predicts the GMAE of diverse adsorption systems without the acquisition of any site-binding information." Please see AdsorbDiff: Adsorbate Placement via Conditional Denoising Diffusion, Kolluru and Kitchin, ICML 2024. They show a method that directly predicts the relaxed structure without having to perform relaxations.

Response: We thank the reviewer for this comment. We recognized that AdsorbDiff represents an important advancement in predicting relaxed adsorption structures. Accordingly, we have removed this sentence from the Introduction section of our manuscript. We have also cited AdsorbDiff in the Introduction and acknowledge its valuable contributions.

5) Reword “We built three GMAE benchmark datasets named OCD-GMAE, Alloy-GMAE and FG-GMAE”. It'd be more accurate to say you "filtered" three datasets.

Response: We thank the reviewer for this comment. We have revised the manuscript, replacing the term "built" with "filtered" or "introduced" where appropriate to more accurately describe our dataset.

Response to Reviews and List of Changes of the Manuscript NCOMMS-24-51449A

Manuscript Number: NCOMMS-24-51449A

Title: A multi-modal transformer for predicting global minimum adsorption energy

Author(s): Junwu Chen, Xu Huang, Cheng Hua, Yulian He and Philippe Schwaller

First of all, we thank the reviewers for their helpful comments. All Reviewers' comments have been taken into account point-by-point. Our detailed response to the Reviewers' comments and the list of changes (marked with highlighting in the revised manuscript) are provided below.

Reviewer #1

I think the authors addressed most of the concerns from the reviewers, and the paper is ready for publication from my point of view. However, I am not an expert on this energy calculation. As an experimentalist, my judgement is based on the method's usefulness for analyzing adsorption energy in a practical system. The opinion of the other two reviewers may be more important.

Response: Thanks a lot for the reviewer's feedback. We are glad that our revisions were satisfactory. We appreciate the reviewer's time and input.

Reviewer #2

The authors have addressed the questions raised by this reviewer and is publishable in its modified form.

Response: Thanks a lot for the reviewer's feedback. We are glad that our revisions were satisfactory. We appreciate the reviewer's time and input.

Reviewer #3

Thank you to the authors for providing a thorough response to the reviewer's comments.

After reading the responses, my main question remains whether the results on FG and Alloy make up for the results on OC20. The results on FG and Alloy demonstrate that with enough training data of similar surfaces and adsorbates, the AdsMT approach can be effective. The 8:1:1 split makes it very likely that the same surface and adsorbate will be in both the training and test sets (but not the exact same pairwise combination). However, if the approach must generalize to unseen surfaces the performance is much worse (OC20 results).

So is an approach that must have training data containing the surface in question useful? I would argue this would severely limit its practical utility. As an example, for the Alloy dataset there are ~6 samples per surface, which would likely result in 5 going to train and 1 going to test. So to get 66% accuracy you need 5 GMAE's on average for a surface in the training data - not a good trade-

off. The paper would be much more convincing if the 8:1:1 split was done to hold out testing surfaces for FG and Alloy (i.e., the 8:1:1 was done on surfaces, not randomly per sample). This would demonstrate whether the approach generalizes to new surfaces outside of training.

Response: Thank you for your valuable feedback regarding model generalization. To address your concern about the practical utility of our approach, we conducted additional experiments testing generalization to unseen surfaces and adsorbates:

For the Alloy-GMAE dataset (1916 surface types), we implemented surface-based splitting: 80% of unique surface types for training, 10% for validation, and 10% for testing. For FG-GMAE (202 adsorbate types), we used adsorbate-based splitting with the same ratios.

Results show our model maintains strong performance even with unseen data:

- Surface-based splitting (Alloy-GMAE): MAE increase of ~ 0.02 eV, success rate decrease of $\sim 6\%$
- Adsorbate-based splitting (FG-GMAE): MAE increase of ~ 0.04 eV, success rate decrease of $\sim 8\%$

Notably, AdsMT still achieves a 60% success rate on unseen surfaces and 65.3% on unseen adsorbates, demonstrating robust generalization capability. These results are now included in Supplementary Information Tables S15-18 and discussed in the main text (lines 179-182).

Table 1: Test MAE (eV) results of AdsMT models using different dataset splitting methods on the Alloy-GMAE dataset.

Split method	AdsMT (SchNet)	AdsMT (eSCN)	AdsMT (GemNet-OC)	AdsMT (ET)	AdsMT (AdsGT)
Random	0.162 ± 0.010	0.163 ± 0.011	0.146 ± 0.010	0.146 ± 0.009	0.143 ± 0.008
Surface type	0.193 ± 0.015	0.176 ± 0.014	0.165 ± 0.018	0.161 ± 0.013	0.158 ± 0.014
Performance change	0.031	0.013	0.019	0.015	0.015

Table 2: Test success rate (SR, %) results of AdsMT models using different dataset splitting methods on the Alloy-GMAE dataset.

Split method	AdsMT (SchNet)	AdsMT (eSCN)	AdsMT (GemNet-OC)	AdsMT (ET)	AdsMT (AdsGT)
Random	57.7 ± 1.7	56.8 ± 2.0	65.0 ± 1.6	64.3 ± 1.7	66.3 ± 1.3

Surface type	48.5 ± 1.9	51.2 ± 1.8	57.2 ± 2.5	58.9 ± 1.7	60.1 ± 2.0
Performance change	-9.2	-5.6	-7.8	-5.4	-6.2

Table 3: Test MAE (eV) results of AdsMT models using different dataset splitting methods on the FG-GMAE dataset.

Split method	AdsMT (SchNet)	AdsMT (eSCN)	AdsMT (GemNet-OC)	AdsMT (ET)	AdsMT (AdsGT)
Random	0.103 ± 0.012	0.102 ± 0.009	0.097 ± 0.007	0.098 ± 0.008	0.095 ± 0.007
Adsorbate type	0.140 ± 0.031	0.147 ± 0.036	0.138 ± 0.038	0.127 ± 0.026	0.123 ± 0.018
Performance change	0.037	0.045	0.041	0.029	0.028

Table 4: Test success rate (SR, %) results of AdsMT models using different dataset splitting methods on the FG-GMAE dataset.

Split method	AdsMT (SchNet)	AdsMT (eSCN)	AdsMT (GemNet-OC)	AdsMT (ET)	AdsMT (AdsGT)
Random	68.9 ± 3.6	69.8 ± 2.9	71.2 ± 2.3	71.6 ± 2.6	71.9 ± 2.4
Adsorbate type	59.1 ± 5.3	58.2 ± 5.7	61.4 ± 4.8	64.3 ± 4.3	65.3 ± 3.8
Performance change	-9.8	-11.6	-9.8	-7.3	-6.6

The Uni-Mol+ help in providing a suitable baseline for the paper. Thank you for these results! The OC20 results should be included in the main text since this would be a likely "naive" approach for tackling the problem (the results on FG and Alloy don't make as much sense since they weren't trained on that data). It demonstrates that the problem is hard, and even these sampling approaches only get ~32% SR.

Response: We thank the reviewer for this comment. We agree that the Uni-Mol+ baseline results provide valuable context and added them to the main text (lines 187-190) and Supplementary Information Table S19.

Response to Reviews and List of Changes of the Manuscript NCOMMS-24-51449C

Manuscript Number: NCOMMS-24-51449C

Title: A multi-modal transformer for predicting global minimum adsorption energy

Author(s): Junwu Chen, Xu Huang, Cheng Hua, Yulian He and Philippe Schwaller

First of all, we thank the reviewers for their helpful comments. All Reviewers' comments have been taken into account point-by-point. Our detailed response to the Reviewers' comments and the list of changes (marked with highlighting in the revised manuscript) are provided below.

Reviewer #3

I thank the reviewers for the additional results supplied in the latest revision. While the success rate decreased with the new splits, they are still reasonable. I would recommend replacing the results in the main manuscript with those using the surface type splits since this offers a more practically useful metric that others may follow.

The authors have addressed my concerns.

Response: Thanks a lot for the reviewer's feedback. The surface splitting results were added to the Section 2.3 of main text:

More challenging dataset splits based on surface or adsorbate type were tested on the Alloy-GMAE and FG-GMAE datasets to explore the generalization performance of AdsMT to unseen surfaces or adsorbates (Supplementary Table 15-18). For surface- or adsorbate-based data partitioning, a set of unique types was obtained, of which 80 % types were randomly sampled for training, and each 10 % types were used for validation and testing, respectively. Therefore, the types of surfaces or adsorbates present in the test set are not included in the training and validation sets. As shown in Supplementary Table 15 and 16, using surface-based data split leads to an increase of approximately 0.02 eV in the MAE with a corresponding decrease of around 6 % in the success rate compared to random split. Although the prediction accuracy slightly decreases under surface-based data partitioning, the best MAE and success rate of AdsMT model on Alloy-GMAE were 0.158 eV and 60.1 %, respectively. Similarly, as presented in Supplementary Table 17 and 18, adsorbate-based data partitioning results in an increase of approximately 0.04 eV in MAE and a reduction of about 8 % in success rate compared to random split, and AdsMT achieves the best MAE of 0.123 eV and the best success rate of 65.3 % on the FG-GMAE. The slight accuracy drops demonstrate the robust generalization capability of AdsMT to unseen surfaces or adsorbates.

We are glad that our revisions were satisfactory. We appreciate the reviewer's time and input.